# Nuclear reassembly defects after mitosis trigger apoptotic and p53-dependent safeguard mechanisms in *Drosophila*

Jingjing Li[1,2], Laia Jordana[1,2], Haytham Mehsen[1,2], Xinyue Wang[1],
Vincent Archambault[1,2]*

**1** Institute for Research in Immunology and Cancer, Université de Montréal, Montreal, Canada,
**2** Département de biochimie et médecine moléculaire, Université de Montréal, Montreal, Canada

\* vincent.archambault.1@umontreal.ca

**Data Availability Statement:** All relevant data are within the paper and its Supporting Information files.

## Abstract

In animals, mitosis involves the breakdown of the nuclear envelope and the sorting of individualized, condensed chromosomes. During mitotic exit, emerging nuclei reassemble a nuclear envelope around a single mass of interconnecting chromosomes. The molecular mechanisms of nuclear reassembly are incompletely understood. Moreover, the cellular and physiological consequences of defects in this process are largely unexplored. Here, we have characterized a mechanism essential for nuclear reassembly in *Drosophila*. We show that Ankle2 promotes the PP2A-dependent recruitment of BAF and Lamin at reassembling nuclei, and that failures in this mechanism result in severe nuclear defects after mitosis. We then took advantage of perturbations in this mechanism to investigate the physiological responses to nuclear reassembly defects during tissue development *in vivo*. Partial depletion of Ankle2, BAF, or Lamin in imaginal wing discs results in wing development defects accompanied by apoptosis. We found that blocking apoptosis strongly enhances developmental defects. Blocking p53 does not prevent apoptosis but enhances defects due to the loss of a cell cycle checkpoint. Our results suggest that apoptotic and p53-dependent responses play a crucial role in safeguarding tissue development in response to sporadic nuclear reassembly defects.

## Introduction

The passage through mitosis requires a profound transformation of the nucleus. In many eukaryotes including humans, the nuclear envelope (NE) is broken, allowing cytoplasmic centrosomes and chromosomes to connect through a bipolar spindle of microtubules [1,2]. In parallel, replicated chromosomes become condensed and individualized, allowing their correct bipolar attachment and the segregation of sister chromatids on the spindle. Nuclear envelopes and the associated lamina then reassemble around the 2 segregated groups of chromosomes that decondense, generating daughter nuclei [3,4]. The reassembly of a nucleus after mitosis is essential for cell viability, cell proliferation, and development. However, errors in nuclear

**Funding:** This work was funded by a Project Grant from the Canadian Institutes of Health Research (CIHR) to VA (175132). JL and HM received studentships from the Fonds de Recherche du Québec – Santé.

**Competing interests:** The authors have declared that no competing interests exist.

**Abbreviations:** BAF, barrier-to-autointegration factor; LEM, Lap2-Emerin-Man1; NE, nuclear envelope; NPC, nuclear pore complex; NR, nuclear reassembly; VRK, vaccinia-related kinase; WT, wild type.

reassembly (NR) can lead to partial structural defects of the nucleus that may include aberrantly shaped nuclei, micronuclei, and an abnormal lamina. Nuclei harboring such flaws may undergo abnormal nucleocytoplasmic exchanges, abnormal gene expression, DNA damage, or disintegration [5,6]. However, how cells and tissues respond to NR defects *in vivo* is still unclear.

The best characterized form of NR defects is micronuclei. They often originate from an imperfectly synchronous segregation of chromosomes in anaphase, or from defects in the cross-bridging of chromosomes in telophase [7,8]. Micronuclei frequently occur in cancer cells, where their NE often breaks [9,10]. Micronuclei compromise the stability of the chromosomes they contain as they are the site of extensive DNA damage [11–13]. This DNA damage can occur by multiple mechanisms resulting from ruptures of the NE or a loss of control of nucleocytoplasmic transport in micronuclei [14]. Exposure to cytoplasmic nucleases such as TREX1, incomplete DNA replication, and the aberrant formation and processing of RNA-DNA hybrids are some of the main mechanisms contributing to micronuclear DNA damage [12,15,16]. Micronuclei were reported to activate a cGAS-STING-dependent innate immune response in mammals [17]. Other studies suggested that micronuclei can alternatively trigger apoptosis [5,18]. Micronuclei were also linked to cellular senescence [19]. In some cases, nuclear defects can be repaired. A ruptured NE can be resealed by the ESCRT-III membrane fusion machinery [6]. Micronuclei can also reintegrate the main nucleus [5]. Other structural NR defects that arise through mechanisms similar to strictly defined micronuclei are likely to result in comparable cellular and physiological consequences [8,14]. The prevalence and impact of the various possible responses to NR defects in the context of tissue development *in vivo* are unclear. Investigating this question requires the ability to experimentally interfere specifically with NR mechanisms in a developing animal.

Nuclear reassembly involves the formation of the lamina, a protein network at the interface between chromatin and the NE in animal cells [20]. The lamina confers structure and rigidity to the nucleus, and it contributes to organizing chromatin domains and regulating transcription [21]. This structure can be composed of A/C-type lamins and B-type lamins. These proteins form dimers that further assemble into higher-order structures [20,22]. Mutations in lamin genes cause congenital diseases referred to as laminopathies that include various forms of progeria and dystrophies [22,23]. At the cellular level, these mutations often cause an irregular lamina and an abnormal nuclear shape. Lamina defects are associated with a propensity of the NE to form blebs and ruptures [24]. Lamina down-regulation and abnormalities are also associated with normal aging and cancer [25–27].

In recent years, the molecular mechanisms that govern NR after mitosis have become better understood. A central pathway in this process involves the protein barrier-to-autointegration factor (BAF) [28]. This dimeric DNA-binding protein is recruited to chromosomes in telophase and links chromosomes to each other as they initiate the reassembly of a single nucleus [7,29–32]. In addition, BAF interacts with Lamin A/C and with transmembrane proteins of the NE that contain a Lap2-Emerin-Man1 (LEM) domain [33–36]. In human cells, the loss of BAF strongly increases their propensity to micronucleation during NR [7]. As cells enter mitosis, the phosphorylation of BAF by vaccinia-related kinases (VRKs) induces the dissociation of BAF from DNA [37,38]. During mitotic exit, the dephosphorylation of BAF is required for its recruitment to chromosomes. This dephosphorylation depends on Protein Phosphatase 2A (PP2A) and its co-factor Ankle2/Lem4 [39,40]. The PP2A-Ankle2-BAF pathway appears to play a conserved role in NR at least in *Homo sapiens* (humans), *Caenorhabditis elegans* (roundworms), and *Danio rerio* (fish) [39,41]. BAF is conserved in *Drosophila melanogaster*, where it is also required for NR [42,43]. In this system, the phosphorylation of BAF by the VRK ortholog Ballchen/NHK-1 is required for its dissociation from DNA in mitosis and female meiosis

[43,44]. We have previously shown that BAF dephosphorylation and recruitment to reassembling nuclei depend in part on PP2A with its B55/Tws regulatory subunit [43]. An Ankle2 ortholog exists in *Drosophila* and was shown to promote asymmetric cell divisions in the developing larval brain [45]. Consistent with this function, hypomorphic mutations *ANKLE2* cause microcephaly in humans [46,47]. However, whether *Drosophila* Ankle2 is required for NR has not been investigated.

Here, we show that Ankle2 promotes BAF recruitment and NR during mitotic exit. Disruption of Ankle2 or BAF leads to defective nuclei with lamina defects and dispersed, abnormally condensed chromatin. We took advantage of these perturbations to examine the cellular and physiological consequences of NR defects in a proliferative tissue *in vivo*. We found that disrupting Ankle2, BAF, or Lamin function in larval imaginal wing discs results in defective NR, apoptosis, and developmental defects in adult wings. Moreover, we found that disrupting apoptosis or a p53-dependent response in this context strongly exacerbates tissue development defects. Thus, these responses play essential roles in promoting the development of a proliferative tissue that incurs frequent structural nuclear defects after mitosis.

## Results

### *Drosophila* Ankle2 promotes BAF dephosphorylation and recruitment during nuclear reassembly after mitosis

In *C. elegans* and human cells, Ankle2 promotes BAF recruitment during NR [39]. Whether Ankle2 functions in a similar manner in *Drosophila* has not been examined. We used *Drosophila* cells in culture to test if Ankle2 is required for BAF recruitment. RNAi depletion of Ankle2 was verified by western blot using newly generated custom antibodies (Fig 1A). We then examined the localization of N-terminally Flag-tagged BAF (Flag-BAF) (Fig 1B). In control cells, Flag-BAF was enriched in the nucleus relative to the cytoplasm, consistent with the previously observed localization of endogenous BAF [42]. We found that this nuclear enrichment was lost after Ankle2 depletion (Fig 1C). Moreover, while Flag-BAF was enriched at the NE relative to the nucleoplasm in control cells, this enrichment was lost after Ankle2 depletion (Fig 1D). Similar results were obtained with GFP-BAF (S1A Fig). To visualize how cells develop these phenotypes after Ankle2 RNAi, we used video microscopy with D-Mel (d.mel-2) cells expressing GFP-BAF and mCherry-Tubulin (Fig 1E and 1F and S1 and S2 Videos). In control cells, GFP-BAF was dispersed in the cytoplasm as cells entered mitosis, but it became strongly enriched on segregated chromosomes in telophase, before being restricted to the nuclear periphery in interphase. This dynamic localization is consistent with previous results [43]. By contrast, in Ankle2-depleted cells, GFP-BAF never became strongly enriched on chromosomes in telophase and formed aggregates at the nuclear periphery at a later stage (Figs 1E and S1B). Importantly, we found that cells needed to go through mitosis to develop the defective nuclear phenotype when Ankle2 was depleted (note that the cell shown in Fig 1E displays GFP-BAF normally localized at the NE before mitosis).

While BAF phosphorylation at conserved N-terminal sites (Ser2, Thr4, and/or Ser5 in *Drosophila*) promotes its release from DNA in mitosis, BAF recruitment to DNA in telophase requires the dephosphorylation of these sites (Fig 1G) [37,43,44]. We observed multiple bands for GFP-BAF on western blot after SDS-PAGE in the presence of Phos-tag (Fig 1H). Mutation of the phosphorylation sites into alanine residues (GFP-BAF³ᴬ) abolished the upper bands, indicating that they correspond to phosphorylated BAF. We found that the depletion of Ankle2 led to an increase in the slow-migrating forms of GFP-BAF. Finally, we found that GFP-BAF³ᴬ remained localized to chromosomes throughout mitosis, even when Ankle2 was depleted (S1B Fig). Altogether, these results indicate that Ankle2 is required for BAF

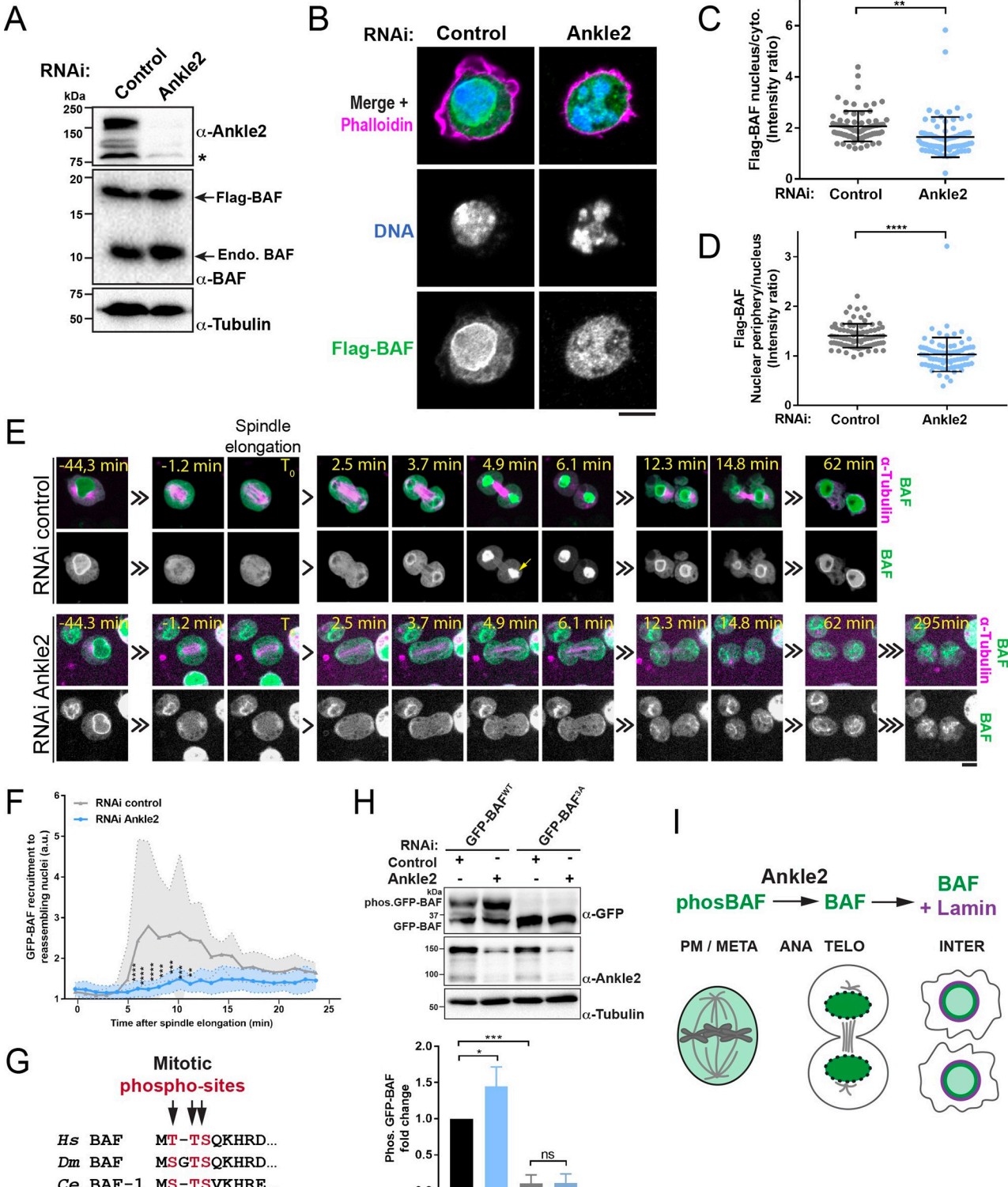

**Fig 1. Ankle2 is required for BAF dephosphorylation and recruitment to reassembling nuclei.** (**A**) Western blots showing the RNAi depletion of Ankle2 in D-Mel cells expressing Flag-BAF. RNAi Control: Non-target dsRNA against bacterial kanamycin resistance gene. *Nonspecific band. The RNAi-sensitive bands likely correspond to Ankle2 splice variants. (**B**) Immunofluorescence showing the subcellular localization of Flag-BAF in cells after RNAi depletion of Ankle2 and in control cells. Cells were also stained with phalloidin to reveal actin. (**C**) Quantification of the nucleus/cytoplasm fluorescence intensity ratios in cells after Ankle2 RNAi or control RNAi. (**D**) Quantification of the nuclear periphery (nuclear envelope)/nucleus

(nucleoplasm) fluorescence intensity ratios in cells after Ankle2 RNAi or control RNAi. For panels C and D, combined results from 3 experiments are shown, where 27 to 30 cells were scored per condition in each experiment. (**E**) Live imaging of mitosis in cells expressing GFP-BAF (green) and mCherry-Tubulin (magenta) after Ankle2 RNAi or control RNAi. $T_0$ was set as the beginning of spindle elongation in anaphase. The strong recruitment of GFP-BAF on DNA in telophase observed in control cells (yellow arrow), does not occur in Ankle2 RNAi cells. All scale bars: 5 μm. (**F**) Quantification of GFP-BAF fluorescence intensity at reassembling nuclei. Fifteen (control RNAi) or 12 (Ankle2 RNAi) cells were quantified. (**G**) Conservation of the mitotic phosphorylation sites in the N-terminus of BAF in humans (*Hs*), *Drosophila* (*Dm*), and *C. elegans* (*Ce*). (**H**) BAF phosphorylation in its N-terminus depends on Ankle2. Cells expressing GFP-BAF WT or 3A (S2A, T4A, S5A) were treated for Ankle2 RNAi or control RNAi. Extracts were submitted to SDS-PAGE with the addition of Phos-tag to increase mobility shifts due to phosphorylation. Western blots for GFP, Ankle2, and Tubulin are shown. Note that slow-migrating bands corresponding to phosphorylated GFP-BAF (phos.GFP-BAF) are eliminated by the 3A mutations and are increased upon Ankle2 RNAi. Fold change shown below blot are the ratios between phos.GFP-BAF and α-Tubulin (loading control). Averages of 4 experiments are shown. All error bars: SD *$p < 0.05$, **$p < 0.01$, ***$p < 0.001$, ****$p < 0.0001$ from paired *t* tests for panel C, D, H and unpaired *t* test for panel F. (**I**) Ankle2 promotes BAF dephosphorylation and recruitment to reassembling nuclei during mitotic exit, as well as BAF-dependent recruitment of Lamin. Coordinate values used to generate graphs are available in S1 Data.

## Ankle2 and BAF promote nuclear reassembly

In human cells, BAF was shown to prevent micronucleation during mitotic exit [7]. BAF is also required for Lamin A recruitment [32,48]. To test if BAF plays a similar role in *Drosophila*, we depleted BAF by RNAi in D-Mel cells and examined nuclear morphology after staining DNA and Lamin ($Dm_0$). To maximize the penetrance of the phenotypes, we transfected dsRNA once or twice, analyzing cells after 4 days or 7 days of depletion, respectively (S2A Fig). We found that BAF depletion results in abnormally shaped or fragmented nuclei and hyper-condensed DNA, when assessed from DAPI staining (Fig 2A–2C). In addition, Lamin was often mislocalized, forming aggregates (Fig 2A and 2D). Thus, BAF is required for normal NR, promoting normal nuclear morphology and Lamin recruitment. Since Ankle2 is required for BAF recruitment, and BAF is required for NR, we tested if Ankle2 was also required for NR. We found that RNAi depletion of Ankle2 induced aberrant nuclei similarly to BAF depletion (Figs 2A–2D and S1A).

To visualize if nuclear defects due to Ankle2 depletion arise during mitosis, we used video microscopy with D-Mel cells expressing H2A-RFP and GFP-Lamin. We found that Ankle2 depletion caused an increase in the emergence of fragmented nuclei at the end of mitosis (S2B and S2C Fig). We previously showed that Lamin recruitment during NR depends on BAF recruitment [43]. Consistent with this notion, we found that Ankle2 depletion led to reduced Lamin recruitment after mitosis (Figs 2E, S2B, and S2C and S3 and S4 Videos). However, depleting Lamin itself caused much less nuclear morphology defects than depleting Ankle2 or BAF (Fig 2A–2D). This is consistent with a Lamin-independent role of BAF in cross-bridging chromosomes during telophase to promote the normal morphology of nascent nuclei [7]. We conclude that Ankle2 is required for the recruitment of BAF and Lamin (Fig 1I) and for normal NR after mitosis in *Drosophila*.

## Depletion of Ankle2 or BAF in wing discs causes nuclear defects and wing development defects

To investigate the cellular and functional consequences of NR defects during tissue development, we used genetic manipulations in imaginal wing discs. These tissues rely on cell proliferation at the larval stage to generate adult wings. To silence the expression of target genes in this tissue, we used the Gal4-UAS system. Nubbin-Gal4 (Nub-Gal4) was used to drive the expression of RNAi constructs under the control of UAS. Nub-Gal4 is expressed specifically in the pouch area of the discs which is destined to develop into the wing proper [49]. We found

dephosphorylation and recruitment to reassembling nuclei in *Drosophila* (Fig 1I), a role that is conserved in *C. elegans* and humans [39].

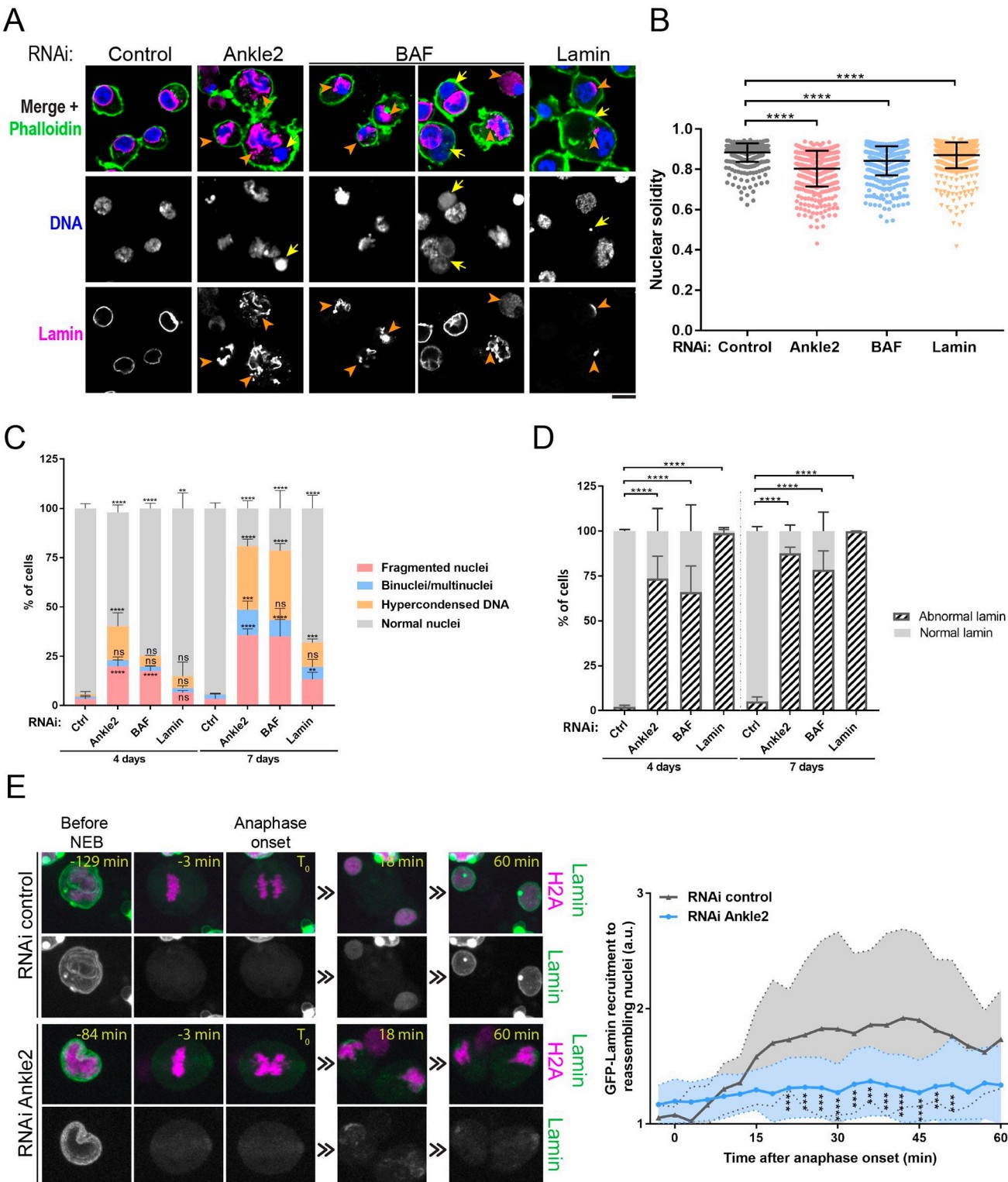

**Fig 2. Ankle2, BAF, and Lamin are required for nuclear reassembly in cells in culture.** (**A**) D-Mel cells were treated by RNAi for 4 days to deplete Ankle2, BAF, or Lamin (or control RNAi). Cells were analyzed by immunofluorescence to reveal Lamin (magenta), DNA (DAPI, blue), and actin (phalloidin, green). After depletion of Ankle2 or BAF, DNA and Lamin are disorganized. After partial depletion of Lamin, cells appear devoid of Lamin or display small patches of Lamin. Yellow arrows: cells with nuclear fragments or binucleated. Orange arrowheads: Lamin aggregates or patches. (**B**) Quantification of nuclear solidity (area/convex surface) based on the DNA staining from cells treated as in A. Combined results from 3 experiments are

shown, where 296 to 442 cells were scored per condition in each experiment. (**C**) Quantification of nuclear phenotypes based on the DNA staining from cells treated by RNAi as indicated. (**D**) Quantification of Lamin phenotypes from cells treated by RNAi as indicated. Abnormal Lamin included aggregated and diffused or absent Lamin. For panels C and D, averages of 3 independent experiments are shown, where 300–400 cells were scored per experiment for each condition. (**E**) Left: Live imaging of mitosis in cells expressing Lamin-GFP (green) and H2A-RFP (magenta) after Ankle2 RNAi or control RNAi. $T_0$ was set as the time of anaphase onset. All scale bars: 5 µm. Right: Quantification of Lamin-GFP fluorescence intensity at reassembling nuclei. Intensities of Lamin-GFP were quantified every 3 min for 12 (control RNAi) and 22 (Ankle2 RNAi) cells. All error bars: SD $**p < 0.01$, $***p < 0.001$, $****p < 0.0001$, ns: nonsignificant from paired $t$ tests for panels B, C, D and unpaired $t$ tests for panel E. Coordinate values used to generate graphs are available in S1 Data.

that depletion of Ankle2 causes nuclear defects in the wing pouch. Immunofluorescence revealed nuclei with irregular lamina surrounding DNA, hypercondensed DNA, masses of DNA devoid of Lamin, and Lamin aggregates separated from DNA (Fig 3A). In parallel to those nuclear defects, adult wings that developed under Nub-Gal4-driven depletion of Ankle2 were smaller than control wings, or almost absent, depending on which RNAi line was used (Figs 3B and S3).

Depletion of BAF using 2 RNAi lines yielded nuclear defects similar to those obtained after Ankle2 RNAi (S4A Fig). The line that resulted in the strongest defects (line #1, VDRC102013) has its UAS RNAi construction inserted at cytolocation 40D which is known to also induce Gal4-dependent overexpression of the transcription factor Tio, which can result in development defects [50]. However, a control line with a UAS insertion at the same site (leading to Tio overexpression only; VDRC60101) did not result in similar nuclear defects (S4A Fig). Thus, nuclear defects observed upon Nub-Gal4-driven depletion of BAF are specifically due to BAF depletion and not Tio overexpression. However, both the BAF RNAi 40D insertion and its Tio-overexpressing control insertion resulted in very small adult wings (S4B and S4C Fig). We conclude that this adult wing phenotype is not specific to BAF depletion for this RNAi line. Nevertheless, depletion of BAF with an alternative RNAi construction (line #2, BDSC36018) resulted in both nuclear defects and reduced adult wing size (S4A–S4C Fig).

Partial depletion of Lamin in wing discs resulted in nuclei lacking Lamin around a large fraction of their periphery, with the remaining Lamin forming foci, similarly to the phenotype observed in cell culture (Fig 3A). This depletion of Lamin also resulted in a reduction of adult wing size, although to a lesser extent than Ankle2 depletion (Fig 3B). Depletion of Ankle2, BAF, or Lamin also caused delocalization of nuclear pore complex (NPC) proteins and Otefin (an inner nuclear membrane protein with a LEM domain that interacts with BAF) from the NE (S4C Fig). Therefore, depletion of Ankle2, BAF, or Lamin in the developing wing causes nuclear structure defects associated with a reduction in adult wing size.

## Genetic validation of an Ankle2-BAF-Lamin pathway required for nuclear reassembly

To identify functional interactions of the Ankle2-BAF-Lamin pathway, we exploited the small wing phenotype. We used an Ankle2 RNAi line (line #1, VDRC100655) which resulted in a partial phenotype due to incomplete penetrance, allowing the identification of enhancer and suppressor genes. To facilitate the identification of enhancer and suppressor genes, we could also modulate the phenotype by varying the temperature because Gal4-UAS-dependent expression increases with temperature. As expected, wing size was further reduced by the introduction of one mutant allele of *baf*, consistent with the role of Ankle2 in promoting BAF recruitment during NR (Fig 3C). Conversely, wing size was partially rescued by introduction of one mutant allele of *ball/nhk-1*, which encodes the kinase that phosphorylates BAF to induce its dissociation from DNA in M-phase [44]. Mutant alleles of *mts* and of *Pp2A-29B*, which encode the catalytic and structural subunits of PP2A, respectively, enhanced the small wing phenotype (Fig 3D). These results are consistent with the role of PP2A in

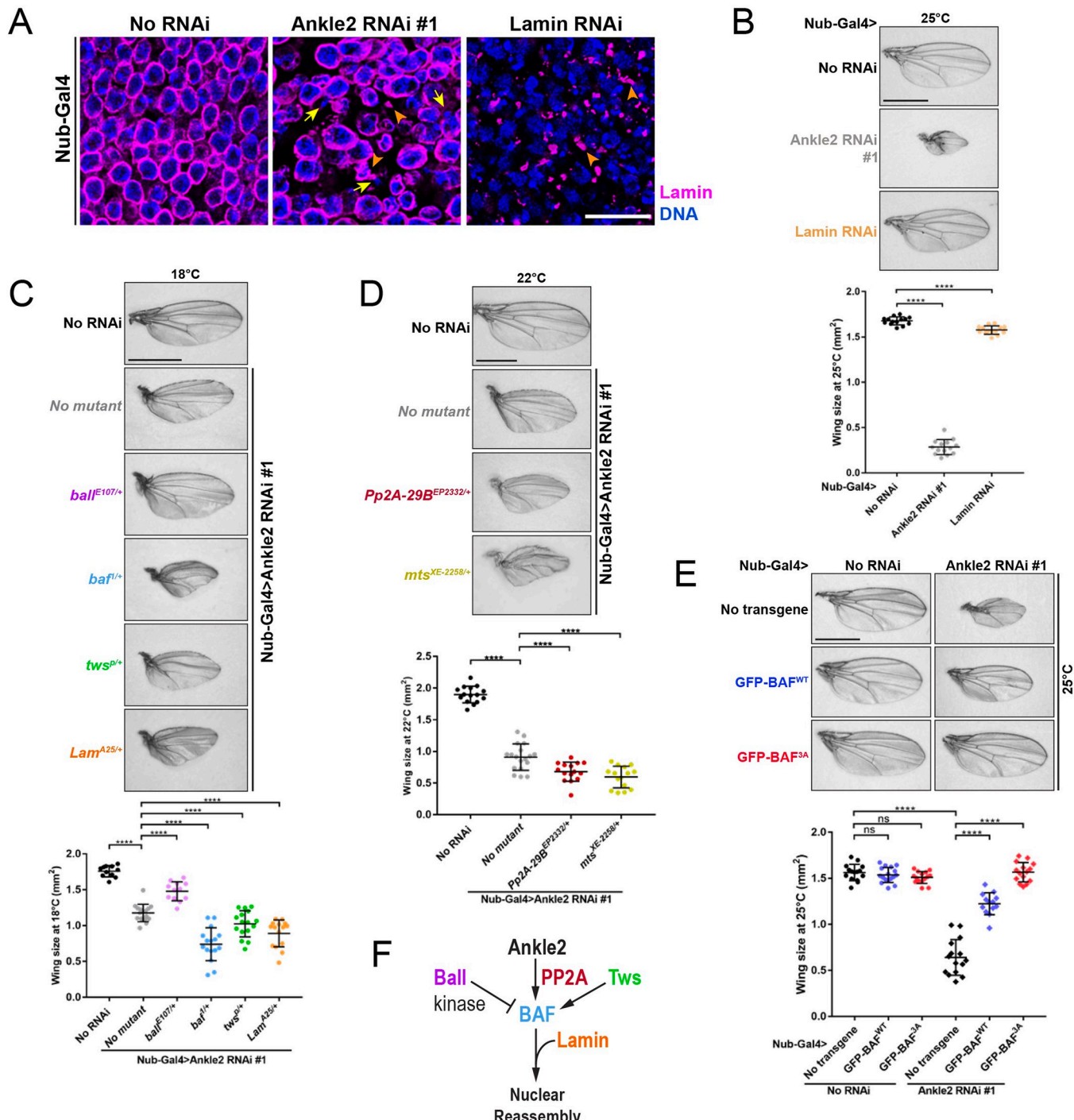

**Fig 3. Disruption of Ankle2 function in BAF regulation during wing development causes nuclear defects and adult wing defects.** (**A**) Nuclear phenotypes in wing discs after induction of Ankle2 RNAi, Lamin RNAi, or no RNAi in the pouch area of the wing disc using Nub-Gal4 at 25°C. Wing discs were analyzed by immunofluorescence against Lamin (magenta) and DNA (DAPI, blue). Yellow arrows: DNA without Lamin. Orange arrowheads: Lamin aggregates or patches. Scale bars: 10 μm. (**B**) Depletion of Ankle2 or Lamin in wing discs results in smaller wings. Top: Examples of adult wings after induction of Ankle2 RNAi, Lamin RNAi, or no RNAi in the pouch area of the wing disc using Nub-Gal4 at 25°C. Bottom: Quantification of wing sizes (surface area, *n* = 14). (**C**) Mutations in *baf*, *tws*, and *lamin* enhance, while a mutation in *ball* suppresses, the small wing phenotype resulting from Ankle2 depletion. Top: Examples of adult wings of the indicated genotypes after induction of Ankle2 RNAi using Nub-Gal4 at 18°C. Bottom: Quantification of wing sizes (*n* = 12 to 16). (**D**) Mutations in PP2A genes enhance the small wing phenotype resulting from Ankle2 depletion. Top: Examples of adult wings of the indicated genotypes after induction of Ankle2 RNAi using Nub-Gal4 at 22°C. Bottom: Quantification of wing sizes (*n* = 12 to 15). (**E**) Expression of GFP-BAF^3A driven by Nub-Gal4 rescues the small wing phenotype due to Ankle2 depletion. Top: Examples of adult wings of the indicated genotypes at 25°C. Bottom: Quantification of

adult wing sizes ($n$ = 10 to 14). In all experiments, Ankle2 depletion was done using line #1 (VDRC100655). All scale bars for adult wings: 1 mm. All error bars: SD ****$p$ < 0.0001, ns: nonsignificant from unpaired $t$ tests with Welch's correction. (**F**) Conceptual representation of the molecular network controlling NR (see text for details). Coordinate values used to generate graphs are available in S1 Data.

dephosphorylating BAF to promote its recruitment during NR, and with the interaction of PP2A with Ankle2 in this process. The small wing phenotype was also enhanced by one mutant allele of *tws*, which encodes the B55 regulatory subunit of PP2A (Fig 3C). This is consistent with our previous finding of a role of PP2A-Tws in the dephosphorylation and recruitment of BAF during NR [43]. Heterozygosity for mutations in *lamin* also enhanced the phenotype (Fig 3C).

To test if wing development defects upon Ankle2 depletion are due to a failure to dephosphorylate BAF, we overexpressed non-phosphorylatable GFP-BAF$^{3A}$. We found that Nub-Gal4-dependent expression GFP-BAF$^{3A}$ in the wing pouch fully rescued wing size when Ankle2 is depleted (Fig 3E). By contrast, expression of GFP-BAF$^{WT}$ rescued wing size only partially. The fact that some rescue is observed suggests that a pool of overexpressed GFP-BAF$^{WT}$ remains unphosphorylated even upon partial depletion of Ankle2. Altogether, these results support the notion that Ankle2 functions with PP2A in promoting the dephosphorylation and recruitment of BAF during NR and that a failure in this function results in developmental defects *in vivo*. Overall, the genetic interactions we observed in the context of wing development are completely consistent with an emerging molecular network where Ankle2 plays a conserved role in NR (Fig 3F).

## Nuclear reassembly defects trigger apoptosis

The nuclear defects observed upon depletion of Ankle2, BAF, or Lamin (fragmented nuclei, hypercondensed DNA, abnormal Lamin, Figs 2 and 3A) are reminiscent of pyknotic nuclei that develop during apoptosis. We therefore sought to determine whether depletion of Ankle2, BAF, or Lamin induces apoptosis. The effector Caspase 3, called Dcp-1 in *Drosophila*, is activated downstream of the canonical apoptosis pathway (Fig 4A) [51,52]. Staining for cleaved, activated Dcp-1 (hereafter Dcp-1) revealed a significant increase in apoptosis after depletion of Ankle2, BAF, or Lamin in the wing disc (Figs 4B, 4C, S5A, and S5B). The intensity of Dcp-1 signals mirrored the penetrance of the small-wing phenotype with 2 Ankle2 RNAi lines (S3 and S6 Figs). An increase in apoptosis was also observed after depletion Otefin or Man1 (2 structural LEM-D proteins of the NE that interact with BAF) or Elys (an NPC component) (Fig 4B and 4C). Moreover, the Dcp-1 signal tended to be located in areas with defective nuclei (S6B Fig). Increased apoptosis was also detected by TUNEL assay in wing discs depleted of Ankle2, BAF, or Lamin (Figs 4D–4F, S5C, and S5D). TUNEL and Dcp-1 signals tended to increase together, and foci were often observed in close proximity suggesting that they tend to occur in the same cells (Fig 4E and 4F). Depletion of Ankle2 or BAF in D-Mel cells in culture also increased Dcp-1 signals concomitantly with hypercondensed DNA in pyknotic nuclei (S7A Fig). Altogether, these results indicate that disruption of NR triggers apoptosis.

To test whether apoptosis was caspase dependent, we used P35, a protein known to prevent apoptosis in *Drosophila* by inhibiting effector caspases [53,54]. Expression of P35 in wing discs depleted of Ankle2, BAF, or Lamin blocked apoptosis measured by TUNEL, as expected (Fig 4D–4F). Interestingly, while expression of P35 eliminated bright Dcp-1 foci, it actually increased a diffuse Dcp-1 signal (Fig 4G and 4H). This is consistent with the fact that while P35 inhibits effector caspases including Dcp-1, it does not block the initiator caspase Dronc, which cleaves Dcp-1 [55,56]. Persistent, diffuse Dcp-1 signals were previously observed following P35 expression [57,58]. In this context, cells positive for Dcp-1 likely accumulate as they do

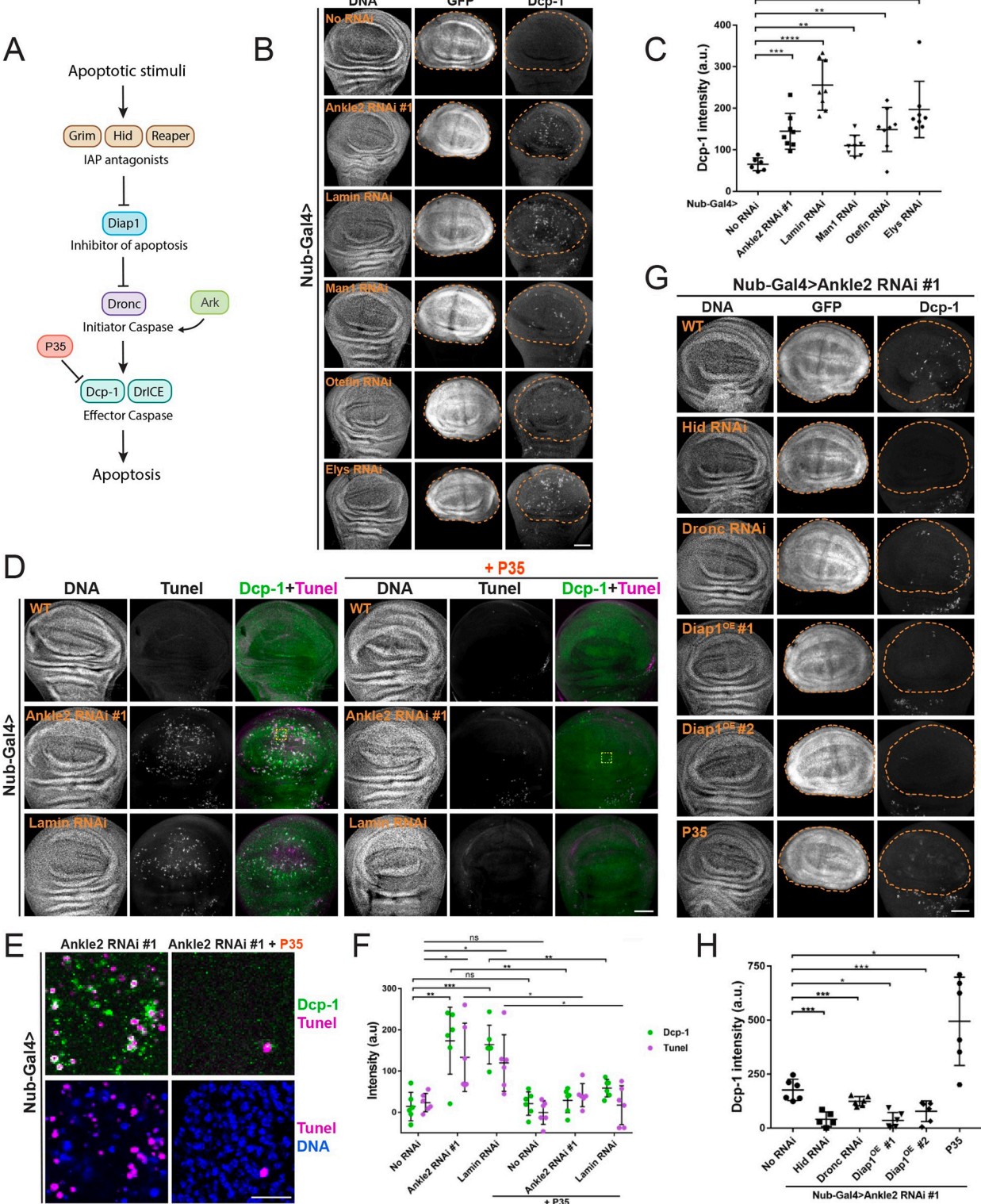

**Fig 4. Perturbation of nuclear reassembly mechanisms induce apoptosis.** (**A**) The classical apoptotic pathway in *Drosophila*. (**B**) Depletion of Ankle2, Lamin, BAF interactors, or Elys induces apoptosis. RNAi for the indicated proteins was induced in the wing pouch by Nub-Gal4 at 25˚C. In parallel, UAS-GFP was used as a marker of the region of interest (wing pouch, inside dotted lines). Wing discs were analyzed by immunofluorescence against cleaved Dcp-1 and stained for DNA (DAPI). (**C**) Quantification of Dcp-1 intensities measured in the wing pouch region of individual wing discs of the indicated genotypes as in B ($n$ = 6 to 8). (**D**) Detection of apoptosis by TUNEL (magenta) and simultaneous

immunofluorescence for cleaved Dcp-1 (green) in wing discs depleted of Ankle2 or Lamin. Expression of P35 (right) abrogates apoptosis. Yellow boxes: Areas enlarged in panel E. (**E**) Enlargements of regions of interest from images in D (yellow boxes). Scale bars: 10 μm. (**F**) Quantifications of TUNEL and Dcp-1 signals in wing discs of the indicated genotypes ($n = 6$). (**G**) Genetic perturbations abrogating apoptosis decrease Dcp-1 signals in the Ankle2-depleted wing pouch region (GFP positive, inside dotted lines). (**H**) Quantification of Dcp-1 intensities measured in the wing pouch of individual wing discs of the indicated genotypes as in G ($n = 6$). Note that a diffuse Dcp-1 signal increases following expression of P35 in Ankle2-depleted wing discs. In all experiments, Ankle2 depletion was done using line #1 (VDRC100655). All scale bars for panels B, D, and G: 50 μm. All error bars: SD $*p < 0.05$, $**p < 0.01$, $***p < 0.001$, $****p < 0.0001$, ns: nonsignificant from unpaired $t$ tests with Welch's correction. Coordinate values used to generate graphs are available in S1 Data.

not complete apoptosis. This increase in diffuse Dcp-1 signal was not detected when this staining was done in combination with TUNEL, for an unknown reason (Fig 4F). As expected, interfering with apoptosis upstream of P35 abrogated Dcp-1 signals; RNAi depletion of Dronc or Hid (2 activators of apoptosis) or overexpression of Diap1 (inhibitor of apoptosis), decreased Dcp-1 signals in Ankle2-depleted wing discs (Fig 4G and 4H).

To further validate that the apoptosis observed upon Ankle2 depletion reflects a failure in the Ankle2-BAF pathway, we tested genetic perturbations predicted to enhance or rescue apoptosis. As expected, a mutant allele of *baf* increased Dcp-1 signals, while a mutant allele of *ball* decreased them (S8A Fig). In addition, overexpression of GFP-BAF$^{WT}$ or GFP-BAF$^{3A}$ decreased Dcp-1 signals (S8B Fig).

The occurrence of apoptosis in wing discs depleted of NR factors raised the possibility that the nuclear defects observed were a consequence (downstream), rather than a cause (upstream) of apoptosis. To test this possibility, we examined cells in wing discs where apoptosis was blocked by the expression of P35. Interestingly, P35 did not prevent the occurrence of defective nuclei in wing discs depleted of Ankle2 or BAF (Fig 5A). In fact, P35 expression appeared to enhance nuclear defects. In addition, only a fraction of Ankle2-depleted D-Mel cells with nuclear defects were Dcp-1 positive (S7B Fig). We conclude that these defective nuclei are not a consequence of apoptosis. Instead, defective NR appears to be the trigger of an apoptotic response that allows the elimination of defective cells during tissue development.

## Apoptosis promotes normal tissue development when nuclear reassembly is compromised

Because apoptosis eliminates cells, we anticipated that blocking apoptosis may partially rescue the small wing phenotypes resulting from depletion of Ankle2, BAF, or Lamin. Strikingly, we obtained the opposite result. Expression of P35 in Ankle2-depleted wing discs enhanced the phenotype, causing adult wings to be even smaller, while expression of P35 alone had no effect on wing size (Fig 5B and 5C). Moreover, overexpressing Diap1 (negative regulator of apoptosis) or RNAi depletion of Hid, Ark, or Dronc (positive regulators of apoptosis) enhanced the small wing phenotype resulting from Ankle2 depletion. Similar results were obtained in the context of the small wing phenotype upon Lamin RNAi (Fig 5D). We conclude that apoptosis promotes normal development when NR defects occur in the wing disc, a proliferative tissue.

## Requirements for apoptosis in response to nuclear reassembly defects

We sought to determine how NR defects trigger apoptosis. NR defects including micronuclei and lamina defects make the NE rupture-prone and can lead to DNA damage [19,59]. Thus, we hypothesized that NR defects may lead to DNA damage that could trigger apoptosis. We found that depletion of Ankle2 in wing discs led to an increase in H2Av phosphorylated at Ser137 (pH2Av), a marker of double-stranded breaks (Fig 6A). It is known that DNA fragmentation can occur as part of the apoptotic process [60]. However, blocking apoptosis by expression of P35 did not eliminate the pH2Av staining (Fig 6A and 6B). These results suggest that DNA damage occurs as a result of NR defects, upstream of apoptosis.

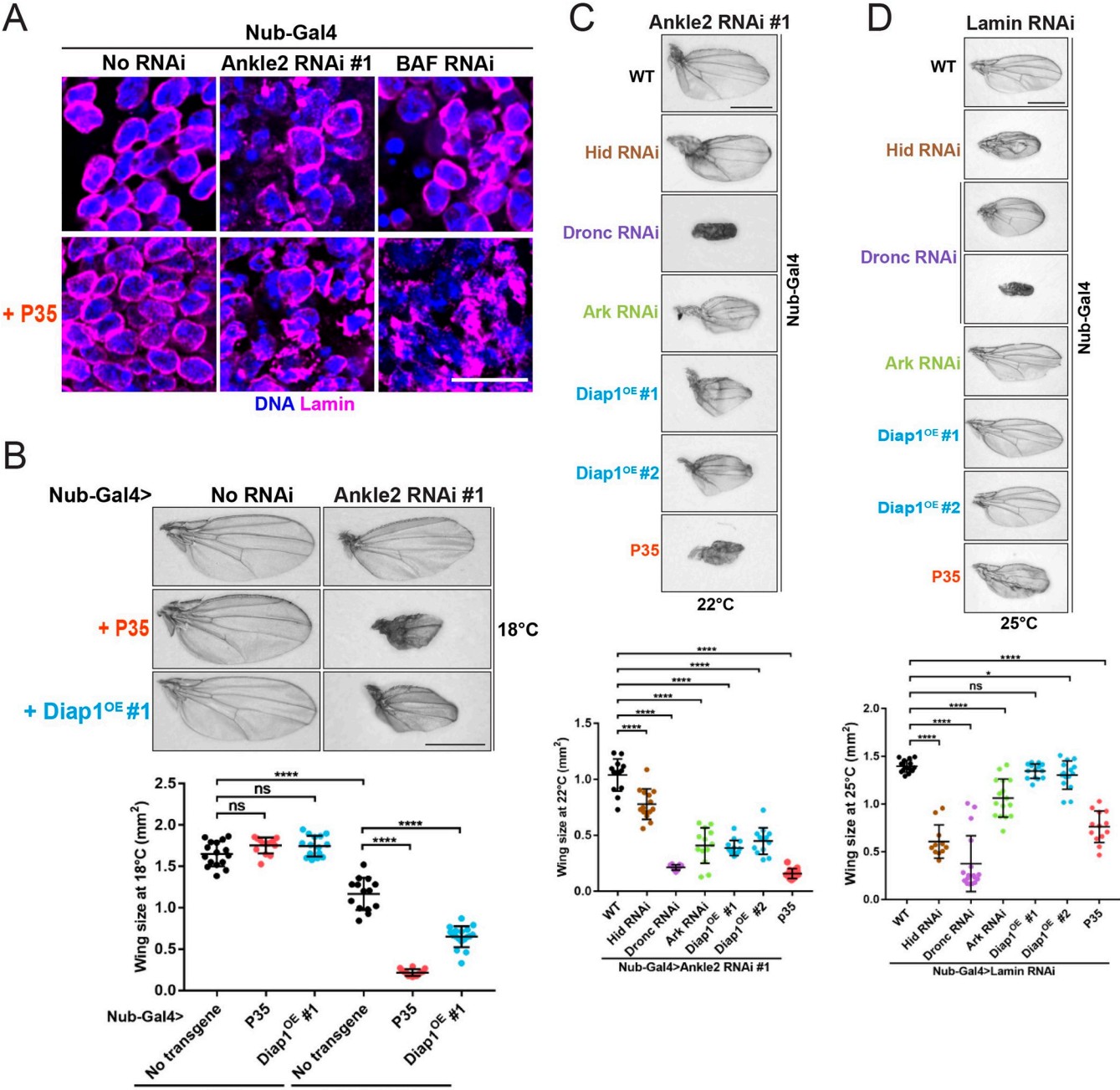

**Fig 5. Apoptosis following nuclear reassembly defects promotes normal tissue development.** (**A**) Expression of P35 in the wing pouch does not eliminate nuclear defects induced by depletion of Ankle2 or BAF. RNAi constructions and P35 expression were driven by Nub-Gal4 at 25°C and wing discs were analyzed by immunofluorescence. Scale bars: 10 μm. (**B**) Expression of P35 or overexpression of Diap1 enhances the small wing phenotype resulting from Ankle2 depletion. Top: Examples of adult wings of the indicated genotypes at 18°C. Bottom: Quantifications of wing size from flies of the indicated genotypes (*n* = 11 to 15). (**C, D**) Depletion of positive regulators of apoptosis (Hid, Dronc, Ark) or expression of negative regulators of apoptosis (Diap1, P35), enhances the small wing phenotype resulting from depletion of Ankle2 (C) or Lamin (D). Top: Examples of adult wings of the indicated genotypes. Bottom: Quantifications of wing size from flies of the indicated genotypes (panel C: *n* = 10 to 11; panel D: *n* = 8 to 14). In all experiments, Ankle2 depletion was done using line #1 (VDRC100655). All scale bars for adult wings: 1 mm. All error bars: SD *$p < 0.05$, ****$p < 0.0001$, ns: nonsignificant from unpaired *t* tests with Welch's correction. Coordinate values used to generate graphs are available in S1 Data.

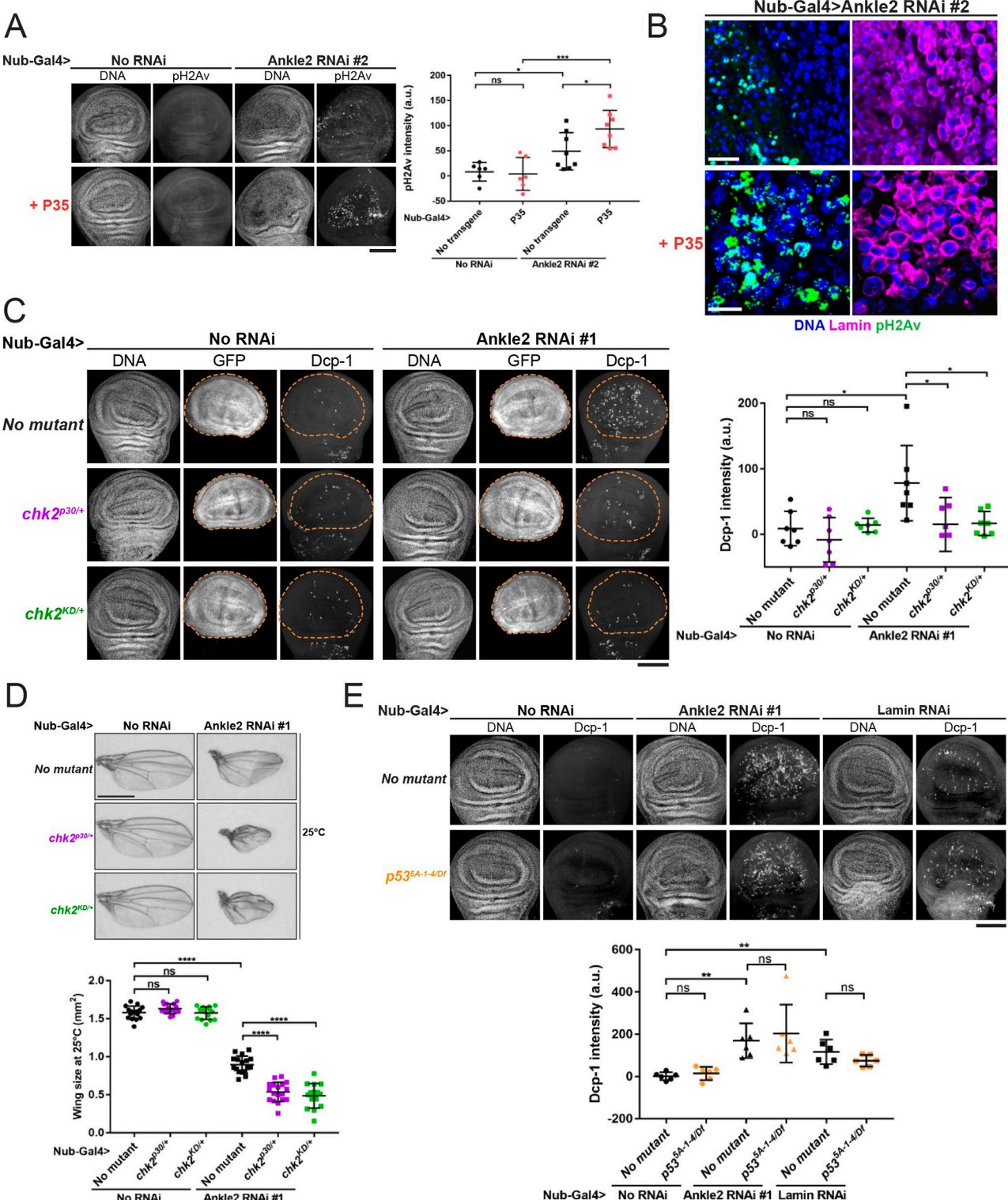

**Fig 6. Apoptosis following nuclear reassembly defects occurs downstream of DNA damage and requires Chk2 but not p53.** (**A**) Depletion of Ankle2 in the wing pouch (line #2, BDSC77437) induces DNA damage independently from apoptosis. Wings discs of the indicated genotypes were stained for pH2Av (pSer137), a marker of DNA double-stranded breaks. Left: Examples of wing discs of indicated genotypes. Right: Quantification of the signal in the entire wing discs ($n$ = 6 to 8). Ankle2 RNAi and P35 expression were driven by Nub-Gal4. (**B**) Images of a higher magnification showing pH2Av associated with DNA in defective nuclei after Ankle2 depletion in the wing pouch. Scale bars: 10 μm. (**C**) Chk2 is required for apoptosis following Ankle2 depletion in the wing pouch. Left: Examples of wing discs of the indicated genotypes. Right: Quantifications of Dcp-1 signals in the wing pouch

(GFP-positive, inside dotted line in images; *n* = 7). (**D**) Mutations in *Chk2* enhance the small wing phenotype resulting from Ankle2 depletion. Top: Example of adult wings of indicated genotype. Scale bar: 1 mm. Bottom: Quantifications of wing sizes at 25°C (*n* = 15 to 18). (**E**) p53 is not required for apoptosis following Ankle2 or Lamin depletion in the wing pouch. Top: Examples of wing discs of the indicated genotypes. Bottom: Quantifications of Dcp-1 signals in the wing pouch (GFP-positive, inside dotted line in images, *n* = 6). In experiments of panels C–E, Ankle2 depletion was done using line #1 (VDRC100655). All scale bars for whole imaginal wing discs: 50 μm. All error bars: SD *$p < 0.05$, **$p < 0.01$, ***$p < 0.001$, ****$p < 0.0001$, ns: nonsignificant from unpaired *t* tests with Welch's correction. Coordinate values used to generate graphs are available in S1 Data.

It was previously shown that nuclear lamina defects due to the loss of BAF or Otefin in *Drosophila* female germline stem cells trigger their death in a manner that depends on the DNA damage response kinases ATR and Chk2 [61,62]. An increase in Dcp-1 signal was observed in female germ cells of an *otefin* mutant or upon RNAi depletion of BAF [62]. Mutation of *chk2* abolished Dcp-1 signals in *otefin* mutant and decreased Dcp-1 signals in BAF-depleted germ cells [62]. We were unable to generate *chk2* null flies with the depletion of Ankle2 in wing discs. However, we found that a single mutant allele of *chk2* was sufficient to decrease Dcp-1 signals (Fig 6C). This is consistent with the previous observation that *chk2* can be haploinsufficient [63]. Moreover, heterozygosity for *chk2* mutations enhanced the small-wing phenotype resulting from Ankle2 depletion (Fig 6D). This is consistent with our finding that blocking apoptosis enhanced this phenotype. Thus, our results suggest that Chk2 contributes to the induction of apoptosis in response to NR defects.

In many contexts, the tumor suppressor p53 promotes apoptosis [64,65]. Moreover, p53 is commonly activated by Chk2 for this function in both flies and vertebrates [66–68]. We therefore tested if p53 is required for apoptosis induced by NR defects. Surprisingly, we found that inactivation of p53 with mutant alleles or RNAi did not block apoptosis in wing discs depleted of Ankle2 or Lamin (Figs 6E and S9). Altogether, our results suggest that NR induces DNA damage and Chk2-dependent apoptosis independently from p53.

## A p53-dependent response promotes tissue development when nuclear reassembly is compromised

Although inactivation of p53 did not prevent apoptosis upon NR defects, we found that it resulted in an enhancement of the small wing phenotype upon depletion of Ankle2 or Lamin (Figs 7A and S10A). In addition to its role in promoting apoptosis, p53 also functions in arresting the cell cycle in response to various cellular defects and insults [65,69,70]. In *Drosophila*, p53 is known to arrest the cell cycle by promoting the ubiquitination and degradation of Cyclin E (CycE) [71,72]. To test if this pathway was at play, we simultaneously depleted CycE and Ankle2 by RNAi in developing wings. Strikingly, depletion of both CycE and Ankle2 resulted in larger wings compared to depletion of Ankle2 alone (Fig 7B). Non-target RNAi transgenes were used as controls to exclude the possibility of a rescue resulting from the dilution of Gal4 between 2 UAS elements. In addition, we found that introduction of one mutant allele of *cycE* also partially rescued wing size upon Ankle2 RNAi (S10B Fig). These results suggest that p53 and CycE function against each other in the context of defects in NR.

It was previously shown that p53 promotes CycE degradation by activating expression of Archipelago (Ago), a Fbxw7 family protein that is part of the SCF complex that promotes ubiquitination and subsequent degradation of CycE [71,72]. Consistent with this mechanism being at play in our system, we found that introduction of a single mutant allele of *ago* strongly enhances the small wing phenotype resulting from Ankle2 depletion (Fig 7C). Interestingly, depletion of CycE or mutation of *Ago* in Ankle2-depleted wing discs did not significantly alter apoptotic levels (Fig 7D), consistent with their implication with p53 in a response distinct from apoptosis.

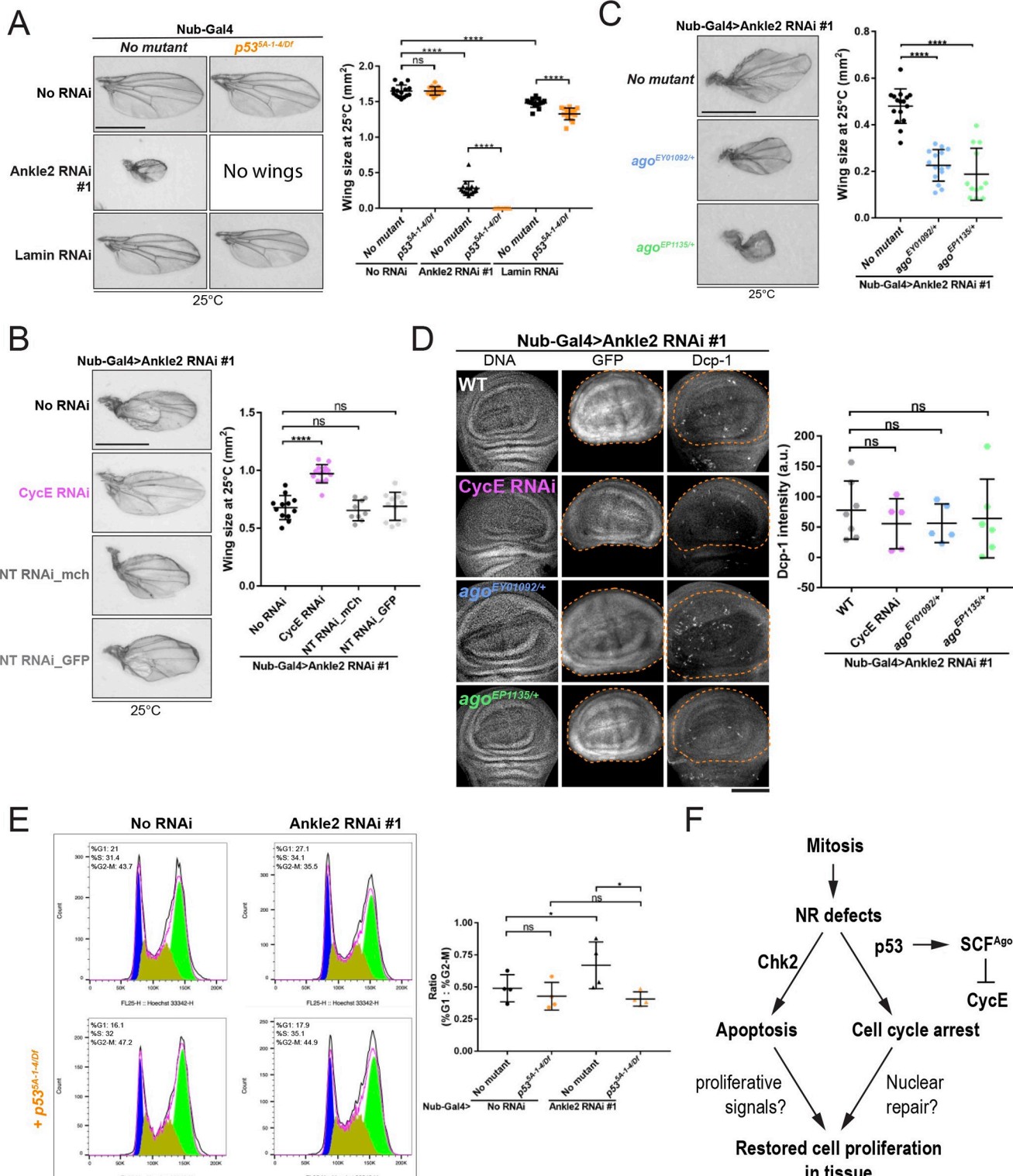

**Fig 7. A p53-dependent response promotes tissue development when nuclear reassembly is compromised.** (**A**) Mutation of *p53* enhances the small wing phenotype resulting from Ankle2 or Lamin depletion. Left: Examples of adult wings of the indicated genotypes. Right: Quantification of wing sizes (*n* = 15 to 16). (**B**) Simultaneous depletion of CycE rescues the small wing phenotype resulting from Ankle2 depletion. Left: Examples of adult wings of the indicated genotypes. Right: Quantification of wing sizes at 25˚C (*n* = 12 to 14). Non-target (NT) RNAi constructions against the mCherry (mch) or GFP sequences were used as controls. (**C**) Mutations in *Ago* enhance the small wing phenotype resulting from Ankle2 depletion. Left: Examples of adult wings of the

indicated genotypes. Right: Quantification of wing sizes at 25˚C (*n* = 12 to 16). (**D**) Depletion of CycE or mutation in *Ago* does not modify apoptosis following Ankle2 depletion. Left: Examples of wings discs of the indicated genotypes. Right: Quantification of the Dcp-1 intensities measured in the wing pouch (GFP-positive, inside dotted line; *n* = 5 to 7). Scale bar: 50 μm. All scale bars for adult wings: 1 mm. All error bars: SD ****$p < 0.0001$, ns: nonsignificant from unpaired *t* tests with Welch's correction. (**E**) Left: Flow cytometry cell cycle analysis (DNA content) of wing disc cells of the pouch area (mCherry positive). Nub-Gal4 was used to drive the depletion of Ankle2 RNAi together with UAS-mCherry, with or without mutation of *p53*. Right: Quantifications of ratios of G1/G2-M cells after Nub-Gal4 depletion of Ankle2 with or without mutation of *p53*. Averages of 4 experiments are shown. Error bars: SD *$p < 0.05$, ns: nonsignificant from paired *t* tests. In all experiments, Ankle2 depletion was done using line #1 (VDRC100655). (**F**) Schematic model showing the p53-dependent cell cycle arrest in response to NR, independently from apoptosis. Both responses may collaborate in restoring normal cell proliferation in a developing tissue. Coordinate values used to generate graphs are available in S1 Data.

Finally, we tested if NR defects in the wing discs induce an alteration of the cell cycle in a p53-dependent manner. Cell cycle profile analysis of wing disc cells by flow cytometry revealed that depletion of Ankle2 increases the population of cells in G1 relative to G2/M (Fig 7E). As expected, mutation of p53 in this context rescued a normal cell cycle profile. Consistent with these observations, a decrease of proliferative cells (EdU+ cells) was also observed in the wing discs upon Ankle2 depletion, and this decreased population of EdU+ cells was also rescued by inactivation of p53 (S10C Fig). These results suggest that in response to NR defects, p53 promotes a cell cycle arrest in G1 via the SCF$^{Ago}$-dependent degradation of Cyclin E, and that this mechanism is independent from apoptosis (Fig 7F).

## Discussion

In this study, we have found that postmitotic NR in *Drosophila* is mediated by a conserved mechanism that involves Ankle2, PP2A, BAF, and Lamin. Using disruptions of this mechanism, we induced NR defects to test the physiological consequences at the cellular and tissue levels. We found that postmitotic NR defects trigger apoptosis. In addition, we identified a parallel mechanism whereby NR defects trigger a p53-dependent cell cycle checkpoint. Our results indicate that both responses are beneficial to tissue development.

Previous work in *C. elegans* and human cells has identified Ankle2/Lem4 as key factor for postmitotic NR [39]. Ankle2 interacts with PP2A and promotes the dephosphorylation of BAF that is required for BAF recruitment on segregated chromosomes and promote NR [39,40]. Recent work on the *Drosophila* ortholog of Ankle2 uncovered its function specifically in the asymmetric divisions of neuroblasts [45]. However, whether *Drosophila* Ankle2 functions in NR was not explored. Our results confirm that Ankle2 does function in a conserved manner to promote BAF dephosphorylation and recruitment to reassembling nuclei. Inactivation of Ankle2 or BAF results in postmitotic NR defects including disorganized DNA and Lamin, which tends to become aggregated and disconnected from DNA. In humans, BAF interacts with Lamin A/C and is required for its recruitment to the lamina [28,73–76]. However, BAF is not required for the recruitment of B-type lamins [76]. *Drosophila* Lamin (Dm$_0$), the only ubiquitously expressed lamin, is considered a closer ortholog to human B-type lamins rather than Lamin A/C [77]. Nevertheless, interfering with BAF function in *Drosophila* results in a failure to recruit Lamin, consistent with the fact that BAF forms a complex with Lamin in *Drosophila* [43,78]. However, we found that while depletion of Lamin leads to NR defects, apoptosis, and developmental defects, these phenotypes tend to be even stronger when Ankle2 or BAF is depleted. This suggests that the known role of BAF in cross-bridging chromosomes in telophase to ensure the formation of a single nucleus of normal shape (independently of lamins) is a crucial, conserved aspect of its functions in NR [7]. In addition, postmitotic nuclear defects observed after inactivation of BAF could be due in part to its proposed function at centromeres, earlier in mitosis [79,80]. Nevertheless, molecular perturbations of nuclear envelope structure and integrity outside the Ankle2-BAF-Lamin pathway can lead to

developmental defects as we have shown that RNAi inactivation of Man1, Otefin, or Elys increases apoptosis. Moreover, mutation of Man1 or RNAi inactivation of Lamin C was also previously shown to result in wing developmental defects [81,82].

Altogether, our results suggest that postmitotic NR defects lead to DNA damage that triggers Chk2-dependent apoptosis. Defective nuclei may harbor holes in their NE or may be more prone to rupture, as demonstrated for micronuclei and nuclei with defective lamina [19,59]. In both cases, DNA would become exposed to cytoplasmic nucleases. The implication of the DNA damage response kinase Chk2 in triggering apoptosis is consistent with previous studies showing that nuclear lamina defects in *Drosophila* female germ cells induce apoptosis through ATR and Chk2 [61,62]. A recent report also showed that a hypomorphic point mutation in *baf* (orthologous to a mutation causing progeria in humans) induced Chk2-dependent apoptosis in *Drosophila* wing discs [80]. Because Chk2 was shown to induce apoptosis through p53 in other contexts, we expected p53 to also be required for apoptosis following NR defects [66–68]. However, we found that p53 is dispensable for this process. Instead, our results suggest that p53 is required for a parallel response where NR-defective cells activate a checkpoint that leads to an arrest or delay in G1. Our genetic observations suggest that the role of p53 in activating SCF$^{Ago}$ for the down-regulation of CycE is at play [71,72]. This p53-dependent response may function to promote the repair of nuclear damage before cells advance in the cell cycle (Fig 7F). Failure in this mechanism would increase cell death, which would be consistent with the enhanced developmental defects we observed.

We discovered that the apoptotic response to sporadic postmitotic NR defects is crucial to tissue development. Strikingly, blocking apoptosis in this context, rather than increasing wing size through the persistence of defective cells, caused a marked decrease in wing size. Previous studies showed that apoptotic cells in proliferating tissues produce signals (Decapentaplegic and Wingless) that induce a compensatory proliferation that promotes tissue homeostasis [83–85]. Therefore, in principle, blocking apoptosis may prevent cells born with defective nuclei from triggering compensatory proliferation to promote the development of a normal-size tissue. However, P35 expression, which we used here to block apoptosis, is known to keep alive cells that initiated apoptotic signaling (termed undead cells), in a state where they overproduce proliferative signals [83,85]. Nevertheless, we observed a decrease rather than an increase in wing size upon P35 expression, excluding the possibility that a lack of compensatory proliferative signals is responsible for smaller wings. In addition, the enhancement of development defects upon P35 expression cannot be attributed to an overproduction of proliferative signals because an enhancement is also observed when apoptosis is blocked at the level of Dronc or upstream (by expression of Diap1 or depletion of Hid), and it is known that the production of proliferative signals requires Dronc [86,87]. Therefore, we favor a model where the persistence of defective cells might negatively impact tissue development. The eventual non-apoptotic death of defective cells may cause damage to the tissue during wing development in a partly non-cell-autonomous manner. It is also possible that these defective cells, before dying, migrate within the tissue, thereby interfering with wing development. It was recently shown that cells with induced chromosomal instability delaminate from the wing disc epithelium and migrate in a manner that depends on caspases and DNA damage [88,89]. Expression of P35 in this context blocked apoptosis but retained enough effector caspase activity to allow cell migration. Our blocking of apoptosis in NR-defective cells could potentially induce a similar response. In any case, our results suggest that the persistence of NR-defective cells is detrimental and that their removal by apoptosis promotes tissue development in a manner that cannot be explained solely on the basis of compensatory proliferation.

In summary, we propose that cells born with structural nuclear defects can follow at least 2 different fates. In some cases, they may repair nuclear damage following a p53-dependent cell

cycle arrest. In other cases, apoptosis functions as a safeguard mechanism that eliminates defective cells and promotes normal tissue development. The relative contributions of these mechanisms in different developmental contexts should be explored. Interestingly, p53 was shown to promote compensatory proliferation following apoptosis in imaginal discs [86], suggesting that both mechanisms may be connected. In addition, it would be interesting to investigate to what extent similar responses function in mammals.

## Materials and methods

### Plasmids, transfections, and cell lines

Plasmids used in this study were generated using the Gateway recombination system (Thermo Fisher Scientific). The cDNA of each gene was cloned into the pDONOR221 vector and sequenced before being recombined into the destination vector downstream of the inducible metallothionein promoter (pMT) or the constitutive Actin 5C promoter (pAc5). The following expression plasmids were generated: pAc5-Flag-BAF, pMT-GFP-BAF, pMT-GFP-BAF³ᴬ, pAc5-mcherry-Tubulin, pAc5-Lamin-GFP, pMT-GFP-Lamin, pMT-RFP-Lamin. Point mutations were generated using QuickChange Lightning Site-Directed Mutagenesis Kit (Agilent, #210518) according to the manufacturer's instructions.

To generate stable cell lines, D-Mel (d.mel-2) cells were transfected using X-tremeGene HP DNA transfection reagent (#06366236001, Roche). Two days after transfection, cells were selected with 20 µg/ml blasticidin for 6 passages. While inducible pMT-based plasmids contained the blasticidin resistance gene, pAc5-based vectors were co-transfected with pCoBlast to render cells resistant to blasticidin. All the cell lines were maintained in the Express Five medium supplemented with penicillin, streptomycin, and glutamine. Cell lines expressing pMT constructs were induced with 300 µm CuS0$_4$ at least 16 h before experiments. Cells expressing GFP-BAF + mCherry-Tubulin, cells expressing H2A-RFP + Lamin-GFP or GFP-Lamin, and cells expressing RFP-Lamin + GFP-BAF (WT or 3A) were described elsewhere [43,90].

For knock-down experiments with RNA interference, dsRNAs were generated from PCR amplicons using a T7 RiboMAX kit (Promega). dsRNA against the bacterial kanamycin resistance gene (KAN) was used as a non-target (NT) control, and 20 µg of dsRNA was transfected into $1 \times 10^6$ D-Mel cells using Transfast transfection reagent (#E2431, Promega) according to the manufacturer's protocol. Cells were then collected and analyzed by immunoblotting, immunofluorescence, or live cell imaging.

### Fly genetics

Flies were cultured in a standard *Drosophila* agar medium at 25˚C. Oregon R was used as wild-type (WT) reference and the fly lines for RNAi were provided by the Bloomington *Drosophila* Stock Center (BDSC) or Vienna Drosophila Resource Center (VDRC). All Fly strains used in this study and their sources are listed in S1 Table. All crosses were also carried out at the indicated temperature (25˚C, 22˚C, or 18˚C) with 60% to 70% humidity. Expression of UAS transgenes in the developing wing pouch was done with the *Nubbin-GAL4* driver (*Nub-Gal4*). The selection of larvae of the desired genotypes was done by avoiding the Tb marker provided on balancer third chromosomes. To select for insertions or mutations on the second chromosomes, the *T(2;3)TSTL14, SM5*: *TM6B Tb¹* pair of balancer second and third chromosomes was used. To facilitate experiments, chromosomes combining *Nub-Gal4*, *Ankle2 RNAi* (VDRC100665) and *Nub-Gal4*, *Lamin RNAi* (VDRC107419) were generated by recombination. To mark the region of interest (wing pouch) for the quantification of Dcp-1 staining and other phenotypes upon *Nub-Gal4*-driven RNAi depletions, *UAS-GFP.nls* (BDSC4776) was

used. To mark cells from the region of interest (wing pouch) for FACS analysis, *UAS-mCherry. nls* (BDSC38425) was used.

## Immunohistochemistry in wing discs

Wandering third instar larvae were dissected in Express Five medium and fixed with 4% paraformaldehyde, 1 mM $CaCl_2$ in PBS for 20 min at room temperature. Following fixation, wing discs were permeabilized and blocked in PBS, 5% BSA, 0.2% Triton for 15 min, and subsequently incubated with primary antibodies overnight at 4˚C. After 3 washes with PBS, 0.2% Triton (PBST 0.2%), wing discs were incubated with secondary antibodies and DAPI in PBST 0.2% with agitation for 1 h at room temperature in the dark. Following 3 washes of 5 min each with PBST 0.2%, wing discs were isolated from the larval tissues and mounted in Mowiol. Primary antibodies used were as follows: anti-cleaved Dcp-1 from rabbit (#9578, Cell signaling Technology. 1:500), anti-Lamin from mouse (Developmental Studies Hybridoma Bank ADL84.12 deposited by P. A. Fisher. 1:500), anti-NPC from mouse (Ab24609, Abcam. 1:200), anti-Otefin from rabbit (custom made by Thermo Fisher Scientific. 1:200). Anti-phospho-Histone H3 (pHH3) from rabbit (EMD Millipore. 1:200), anti-Histone pH2AvD from rabbit (#601-401-914S, Rockland. 1:200). Secondary antibodies used were purchased from Thermo Fisher Scientific and were as follows: Alexa Fluor 555 anti-rabbit (1:200), Alexa Fluor 647 anti-mouse (1:200).

## Imaginal wing disc TUNEL staining

Terminal deoxynucleotidyl transferase dUTP nick end labeling (TUNEL) assay in imaginal wing discs was performed using the ApopTag red in situ apoptosis detection Kit (#S7165, Sigma-Aldrich), following the manufacturer's procedure. Briefly, dissected larval wing discs were fixed in 4% paraformaldehyde in PBS for 20 min at room temperature, then washed twice in PBST 0.2% for 5 min. A postfixation was done in precooled ethanol: acetic acid 2:1 for 5 min at −20˚C. After 3 washes with PBST 0.2%, larval tissues were incubated in 75 μl of Equilibration buffer (90416) for 10 min at room temperature. After completely removing the Equilibration buffer, 55 μl of freshly prepared Working strength TdT reaction buffer was added into larval tissues and incubated for 1 h at 37˚C in a humidified chamber. Shortly after applying stop/wash buffer (90419) and 3 washes with PBST 0.2%, tissues were incubated in the dark with anti-Digoxigenin conjugate working buffer (53% v/v Blocking solution (90425) and 47% v/v anti-digoxigenin conjugate (90429)) with DAPI for 30 min at room temperature and mounted in Mowiol. For TdT end- and Dcp-1 double-labeling, the anti-Dcp-1 antibody was prepared with the blocking solution (90425). Blocking and primary antibody staining were carried out as described for the immunohistochemistry after applying stop/wash buffer. The secondary antibody was diluted with the anti-Digoxigenin conjugate working buffer.

## Imaginal wing disc EdU labeling

Imaginal wing discs from third instar larvae were dissected in Express Five medium at room temperature and incubated for 1 h with 10 μm EdU. Wing discs were then fixed and stained, according to standard protocol (Click-iT EdU cell proliferation Kit for imaging, Alexa Fluor 488, Thermo Fisher Scientific, #C10337). After removing the cocktail reaction buffer, 3 short washes were done and additional staining was applied.

## Western blotting

For western blots, cells were resuspended in PBS containing protease inhibitors and 1 volume of 2X Laemmli buffer (S3401-10VL, Sigma) was added before heating at 95˚C for 2 min.

Extracts were then electrophoresed on 12% Tris-acrylamide SDS-PAGE gel and blotted onto a PVDF membrane (#1620177, Bio-Rad). Membranes were blocked with 5% milk solution for 1 h at room temperature and probed with primary antibody overnight at 4°C. Peroxidase-conjugated anti-rabbit secondary antibodies from goat (1:5,000, 111-035-008, Jackson ImmunoResearch) or anti-mouse from goat (1:5,000, 115-035-003, Jackson ImmunoResearch) were used for primary antibody detection. To visualize protein phosphorylation levels, Phos-tag (#300–93523, FUJIFILM Wako Chemicals) was added into 15% Tris-acrylamide SDS-PAGE gel. After electrophoresis, the gel was soaked in a transfer buffer with 1 mM EDTA for 10 min and then in a transfer buffer without EDTA for another 10 min before being transferred onto a PVDF membrane. Primary antibodies used for WB were as follows: anti-α-Tubulin DM1A from mouse (#T6199, Sigma. 1:5,000), anti-Lamin Dm0 from mouse (Developmental Studies Hybridoma Bank ADL84.12 deposited by P. A. Fisher. 1:1,000), anti-BAF (custom made by Thermo Scientific. 1:1,000), anti-Ankle2 (custom made by Thermo Fisher Scientific. 1:1,000), anti-GFP from rabbit (TP401, Torrey Pines. 1:5,000). The blots were visualized using Clarity Western ECL substrate (170–5060, Bio-Rad) by the ChemiDoc XRS+ system (Bio-Rad). Uncropped images of western blots used in the figures are shown in the Supplementary file S1 Raw Images.

### Immunofluorescence in cells in culture

For immunofluorescence, D-Mel Cells were washed 2 times with PBS before fixation on coverslips with 4% formaldehyde for 20 min at room temperature. After fixation, cells were permeabilized and blocked in PBS containing 0.1% triton X-100 and 1% BSA (PBSBT) for 15 min. Cells were then incubated with primary antibodies diluted in PBSBT for 2 h at room temperature, washed 3 times in PBS containing 0.1% triton X-100 (PBST) and incubated with secondary antibodies and DAPI diluted in PBSBT for 1 h at room temperature. Coverslips were washed 3 times in PBST before being mounted in Vectashield medium (Vector). Primary antibodies used were as follows: anti-Flag from mouse (#F1804, Sigma-Aldrich. 1:2,000), anti-Lamin Dm0 from mouse (Developmental Studies Hybridoma Bank ADL84.12 deposited by P. A. Fisher. 1:500), Alexa Fluor 488 phalloidin (#A12379, Invitrogen. 1:1,000), anti-Dcp-1 from rabbit (#9578, Cell Signaling Technology. 1:200). Secondary antibodies used were purchased from Thermo Fisher Scientific and were as follows: Alexa Fluor 555 anti-rabbit (1:200), Alexa Fluor 488 anti-rabbit (1:200), Alexa Fluor 488 anti-mouse (1:200), Alexa Fluor 647 anti-mouse (1:200).

### Microscopy and quantification

Live imaging in D-Mel cells was performed using a Spinning-disk confocal system (Yokogawa CSU-X1 5000) mounted on a fluorescence microscope (Zeiss Axio Observer Z1). Cells were plated into a LabTek II chambered coverglass (#155409, Thermo Fisher Scientific) for at least 2 h before filming. To monitor GFP-BAF and Lamin-GFP levels over time, GFP signals for each time point at the reassembling nuclei (circular areas) were quantified directly with the Zen software.

All images of imaginal wing discs were taken using the scanning confocal microscope Zeiss Leica SP8. For each imaginal wing disc, 15 focal planes were taken at 0.35 μm intervals and analyzed using Fiji software. Six to 8 wing discs in each condition were used for the quantification of Dcp-1 staining. Briefly, a Z-sum intensity projection was generated for each image and the GFP positive area was defined as the region of interest. Three regions out of the zone of interest in the disc were randomly chosen and the average of the mean intensities of these zones was calculated as background. The specific intensity of Dcp-1 in the region of interest was obtained by subtraction of the background.

The scanning confocal microscope Zeiss LSM880 was used to take images of fixed D-Mel cells and cells from fixed imaginal wing discs (higher magnification than entire discs). Multiple focal planes at 0.16 μm intervals were obtained and subsequently treated with AiryScan software. For quantifications of fixed cells, the mean intensity of fluorescence was measured directly with ZEN software in a single in-focus plane to monitor Flag-BAF fluorescence in the nucleus, the cytoplasm, or at the nuclear periphery. For the nucleus and the cytoplasm, measurements were taken from representative areas. For the nuclear periphery, measurements were taken from the entire area immediately surrounding the DAPI staining. To quantify the number of cells with nuclear phenotypes of interest, maximum-intensity projections were produced from Z-stacks taken with 0.3 μm interspace. Cells with the phenotypes of interest were labeled by hand and the cell counter tool in Fiji software was used to obtain cell numbers. Abnormal Lamin (quantified in Fig 2D) was defined as a strongly irregular or absent Lamin staining around DNA. Nuclear solidity (quantified in Fig 2B) is defined as the ratio of the measured area of the nucleus (based on DAPI staining) to that of a bounding convex shape. Measurement of nuclear solidity is performed using Fiji software.

All images of adult wings were taken using a stereo microscope. Ten to 15 wings from adult females were analyzed for each condition using Fiji software, and the wing size was quantified using the area measurement tool.

### Fluorescence activated cell sorting

At least 30 wing discs from third instar larvae were dissected in Express 5 medium at room temperature and washed twice for 3 min in PBS. Cells were dissociated by incubating wing discs in 10x trypsin-EDTA solution (#T4174, Sigma-Aldrich) in the presence of the DNA dye Hoechst 33342 (#10337G, Invitrogen, 0,5 μg/ml) for 3 h with gentle agitation. Cells were filtered and analyzed using a Yeti Cell Analyzer and Flowjo software.

### Supporting information

**S1 Fig. BAF dynamic localization during mitosis depends on its phosphorylation sites and on Ankle2.** (**A**) Immunofluorescence showing the subcellular localization of GFP-BAF in D-Mel cells after RNAi depletion of Ankle2 and in control cells. Cells were stained for Lamin and with phalloidin to reveal actin. (**B**) Live imaging of mitosis in cells expressing GFP-BAF$^{WT}$ or GFP-BAF$^{3A}$ (green) and RFP-Lamin (magenta) after Ankle2 RNAi or Control RNAi. $T_0$ was set as the beginning of spindle elongation in anaphase. Note that GFP-BAF$^{3A}$ stays on chromosomes throughout mitosis even when Ankle2 is depleted. In addition, the recruitment of RFP-Lamin at reassembling nuclei is rescued. Scale bars: 5 μm. (TIF)

**S2 Fig. Complement to Fig 2—Ankle2, BAF, and Lamin are required for nuclear reassembly in cells in culture.** (**A**) Western blots showing RNAi depletion of Ankle2, BAF, and Lamin 4 or 7 days post-transfection. (**B**) Depletion of Ankle2 causes postmitotic fragmented nuclei. D-Mel cells expressing H2A-RFP and GFP-Lamin were RNAi-treated as indicated dsRNA for 4 days and mitoses were filmed. Arrows: a lagging chromosome in telophase becomes a micronucleus. Scale bar: 5 μm. (C) Quantification of nuclear defects observed by live imaging as in B. Averages of 4 experiments are shown, where 91 and 110 diving cells in total were scored for dsRNA control and dsRNA Ankle2, respectively. $^*p < 0.05$, $^{**}p < 0.01$, ns: nonsignificant from unpaired $t$ tests. Coordinate values used to generate graph are available in S1 Data. (TIF)

**S3 Fig. Wing development defects caused by the depletion of Ankle2 using 2 RNAi.** Ankle2 RNAi was driven by Nub-Gal4 at 25˚C using lines VDRC100655 (Ankle2 RNAi line #1) and BDSC77437 (Ankle2 RNAi line #2). (**A**) Example images of adult flies. Scale bar: 1 mm. (**B**) Quantification of wing sizes at 25˚C ($n = 15$). ****$p < 0.0001$ from unpaired $t$ tests with Welch's correction. Coordinate values used to generate graph are available in S1 Data.
(TIF)

**S4 Fig. Complement to Fig 3—RNAi Depletion of Ankle2, BAF, or Lamin during wing development causes nuclear defects and adult wing defects.** (**A**) Left: Nuclear defects in wing discs resulting from driving BAF RNAi from 2 lines (line #1: VDRC103013; line #2: BDSC36108) with Nub-Gal4 at 25˚C. Both BAF RNAi constructions induce Lamin mislocalization and DNA foci devoid of Lamin. Line #1 (VDRC103013) has the BAF RNAi construction inserted at cytolocation 40D which was shown to also lead to overexpression of Tio. A control line with a UAS element alone inserted at the same site (VDRC60101, Tio$^{OE}$) does not result in similar nuclear defects. Therefore, these defects are specific to BAF depletion for both BAF RNAi lines. However, the Tio$^{OE}$ line results in some level of apoptosis revealed by Dcp-1 staining. Right: Lamin RNAi also induces apoptosis as indicated by Dcp-1 staining. Results from separate experiments are shown in the left and right parts. Scale bar: 10 μm. (**B**) Adult wing defects resulting from driving the 2 BAF RNAi insertions with Nub-Gal4 at 25˚C. Top: Examples of adult wings of the indicated genotypes. Line #1 (VDRC103013) results in a more pronounced phenotype but the control Tio$^{OE}$ line results in a similar small wing phenotype. Therefore, the adult wing phenotype with line #1 is not specific to BAF depletion. Scale bar: 1 mm. Bottom: Quantification of wing sizes at 25˚C ($n = 12$). **$p < 0.01$, ****$p < 0.0001$, ns: nonsignificant from unpaired $t$ tests with Welch's correction. (C) RNAi Depletion of Ankle2 (line #1, VDRC100655), BAF (line #1, VDRC103013) or Lamin results in mislocalization of Otefin and NPC proteins. All constructions were driven with Nub-Gal4 at 25˚C and wings discs were analyzed by immunofluorescence. Scale bar: 5 μm. Coordinate values used to generate graph are available in S1 Data.
(TIF)

**S5 Fig. Complement to Fig 4—RNAi depletion of Ankle2, BAF, or Lamin causes apoptosis.** (**A**) The indicated RNAi constructions were induced in the wing pouch by Nub-Gal4 at 25˚C. In parallel, UAS-GFP was used as a marker of the region of interest (wing pouch, dotted lines). Wing discs were analyzed by immunofluorescence against cleaved Dcp-1 and stained for DNA (DAPI). (**B**) Quantification of the Dcp-1 intensities measured in the wing pouch region of individual wing discs of the indicated genotypes as in A ($n = 6$). (**C**) Detection of apoptosis by TUNEL (magenta) and simultaneous immunofluorescence for cleaved Dcp-1 (green) in wing discs depleted of BAF. Expression of P35 abrogates apoptosis. (**D**) Quantifications of TUNEL and Dcp-1 signals in wing discs of the indicated genotypes ($n = 6$). In all panels: BAF RNAi #1: VDRC103013; BAF RNAi #2: BDSC36108; Ankle2 RNAi #1: VDRC100655. All Scale bars: 50 μm. All error bars: SD *$p < 0.05$, **$p < 0.01$, *** $p < 0.001$, ns: nonsignificant from unpaired $t$ tests with Welch's correction. Coordinate values used to generate graphs are available in S1 Data.
(TIF)

**S6 Fig. Complement to Fig 4—RNAi depletion of Ankle2 using 2 RNAi constructions causes apoptosis in wing discs.** (**A**) Ankle2 RNAi was driven by Nub-Gal4 at 25˚C using lines VDRC100655 (Ankle2 RNAi line #1) and BDSC77437 (Ankle2 RNAi line #2). Top: Cleaved Dcp-1 staining in wing discs of the indicated genotypes. Bottom: Quantification of Dcp-1 signals ($n = 6$). All error bars: SD *$p < 0.05$, **$p < 0.01$, from unpaired $t$ tests with Welch's

correction. (**B**) Nuclear defects and apoptosis are revealed by immunofluorescence against Lamin, DAPI, and cleaved Dcp-1. Arrowheads: hypercondensed DNA in apoptotic cells. Scale bar: 10 μm. (**C**) phospho-Histone H3 (pHH3) staining suggests compensatory cell proliferation around the area of the wing disc where Ankle2 is depleted (line #2, BDSC77437) and apoptosis occurs (detected by TUNEL). All scale bars for imaginal wing discs: 50 μm. Coordinate values used to generate graph are available in S1 Data.
(TIF)

**S7 Fig. RNAi depletion or Ankle2 or BAF results in apoptosis in D-Mel cells in culture.** (**A**) D-Mel cells were transfected with the indicated dsRNA and analyzed by immunofluorescence after 4 days. Top: Examples of images. Arrows: Apoptotic cells with hypercondensed DNA. Scale bar: 20 μm. Bottom: Quantification of Dcp-1 positive (Dcp-1+) cells of the indicated conditions. Averages of 3 experiments are shown, where 292 to 395 cells were scored per experiment in each condition. (**B**) Top: Apoptosis is observed in a fraction of nuclear-defective cells. D-Mel cells were RNAi-treated as indicated and analyzed by immunofluorescence after 4 days. Arrows: Examples of cells with nuclear defects that are Dcp-1-negative. Bottom: Ankle2-depleted cells with Lamin defects were scored for Dcp-1 staining. Averages of 3 experiments are shown, where 256 to 333 cells were scored per condition in each experiment. All error bars: SD **$p < 0.01$, ****$p < 0.0001$, ns: nonsignificant from paired $t$ tests. Coordinate values used to generate graphs are available in S1 Data.
(TIF)

**S8 Fig. Disruption of Ankle2 function in BAF regulation during wing development causes apoptosis.** (**A**) Mutation in *baf* enhances, while mutation in *ball* suppresses, apoptosis resulting from Ankle2 depletion. Left: Examples of wing discs of the indicated genotypes after induction of Ankle2 RNAi using Nub-Gal4 at 25˚C. In parallel, UAS-GFP was used as a marker of the region of interest (wing pouch, inside dotted line). Right: Quantification of Dcp-1 signals in the wing pouch ($n$ = 5 to 6). (**B**) Overexpression of GFP-BAF$^{3A}$ or GFP-BAF$^{WT}$ rescues apoptosis resulting from Ankle2 depletion. Top: Examples of wing discs of the indicated genotypes at 25˚C. Bottom: Quantification of Dcp-1 signals ($n$ = 5 to 6). Analysis was done as in A. In all experiments, Ankle2 depletion was done using line #1 (VDRC100655). All scale bars: 50 μm. All error bars: SD **$p < 0.01$, ****$p < 0.0001$, from unpaired $t$ tests with Welch's correction. Coordinate values used to generate graphs are available in S1 Data.
(TIF)

**S9 Fig. Complement to Fig 6—p53 is not required for apoptosis in response to nuclear reassembly defects.** (**A**) Depletion of p53 does not block apoptosis resulting from Ankle2 depletion (line VDRC100655). RNAi depletions of p53 were achieved with lines VDRC38235 (p53 RNAi #1) and VDRC10692 (p53 RNAi #2). Left: Examples of wing discs of the indicated genotypes after induction of Ankle2 RNAi using Nub-Gal4 at 25˚C, UAS-GFP is used as a marker for the region of interest (pouch area, inside dotted line). Right: Quantifications of Dcp-1 signals in the wing pouch ($n$ = 7 to 8). Scale bar: 50 μm. All error bars: SD **$p < 0.01$, ns: nonsignificant from unpaired $t$ tests with Welch's correction. Coordinate values used to generate graph are available in S1 Data.
(TIF)

**S10 Fig. Complement to Fig 7—A p53-dependent response promotes tissue development when nuclear reassembly is compromised.** (**A**) Depletions of p53 enhance the small wing phenotype resulting from Ankle2 depletion. Left: Examples of adult wings of the indicated genotypes after inducing RNAi using Nub-Gal4 at 25˚C. Right: Quantifications of wing sizes at 25˚C ($n$ = 15 to 18). Scale bar: 1 mm. (**B**) Mutation in *cycE* enhance the small wing

phenotype resulting from Ankle2 depletion. Left: Examples of adult wings of the indicated genotypes at 22°C. Right: Quantifications of wing sizes ($n$ = 13 to 15). Scale bar: 1 mm. In panels A and B, Ankle2 RNAi line #1 (VDRC100655) was used. All error bars: SD **$p < 0.01$, ***$p < 0.001$, ****$p < 0.0001$, ns: nonsignificant from unpaired $t$ tests with Welch's correction. (**C**) A decrease in the population of EdU+ cells resulting from Ankle2 depletion (line #2, BDSC77437) in wing discs is rescued by inactivation of *p53*. Wings discs of indicated genotypes were incubated with EdU for 1 h and then co-stained for DNA (DAPI) and cleaved Dcp-1. Scale bar: 50 μm. Coordinate values used to generate graphs are available in S1 Data. (TIF)

**S1 Video. Complement to Fig 1E—Mitosis in a cell expressing GFP-BAF (green) and mCherry-Tubulin (magenta) after control RNAi.**
(MOV)

**S2 Video. Complement to Fig 1E—Mitosis in a cell expressing GFP-BAF (green) and mCherry-Tubulin (magenta) after Ankle2 RNAi.**
(MOV)

**S3 Video. Complement to Fig 2E—Mitosis in a cell expressing Lamin-GFP (green) and H2A-RFP (magenta) after control RNAi.**
(MOV)

**S4 Video. Complement to Fig 2E—Mitosis in a cell expressing Lamin-GFP (green) and H2A-RFP (magenta) after Ankle2 RNAi.**
(MOV)

**S1 Table. *Drosophila* strains used in this study.** The transgenes, mutant alleles, sources, and complete genotypes are indicated.
(XLSX)

**S1 Data. Numerical data underlying graphs in Figs 1C, 1D, 1F, 1H, 2B, 2C, 2D, 2E, 3B, 3C, 3D, 3E, 4C, 4F, 4H, 5B, 5C, 5D, 6A, 6C, 6E, 7A, 7B, 7C, 7D, 7E, S2C, S3B, S4B, S5B, S5D, S6A, S7A, S7B, S8A, S8B, S9, S10A, and S10B.**
(XLSX)

**S1 Raw Images. Uncropped images of western blots used in the figures.**
(PDF)

# Acknowledgments

We thank Christian Charbonneau and Annie Gosselin for their precious help with the microscopy and FACS, respectively. We thank Caroline Baril and David Hipfner for sharing fly lines, for technical help, and for helpful discussions. We thank Pamela Geyer, Talila Volk, and Nam-Sung Moon for sharing fly lines.

# Author Contributions

**Conceptualization:** Jingjing Li, Vincent Archambault.

**Formal analysis:** Jingjing Li.

**Funding acquisition:** Jingjing Li, Haytham Mehsen, Vincent Archambault.

**Investigation:** Jingjing Li, Laia Jordana, Haytham Mehsen, Xinyue Wang, Vincent Archambault.

**Methodology:** Jingjing Li, Vincent Archambault.

**Project administration:** Vincent Archambault.

**Resources:** Vincent Archambault.

**Supervision:** Vincent Archambault.

**Writing – original draft:** Jingjing Li, Vincent Archambault.

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
