## [Editor Report · Decision Letter 0]

9 Dec 2023

Dear Dr Archambault, 

Thank you for submitting your manuscript entitled "Nuclear reassembly defects after mitosis trigger apoptotic and p53-dependent safeguard mechanisms in Drosophila" for consideration as a Research Article by PLOS Biology.

Your manuscript has now been evaluated by the PLOS Biology editorial staff as well as by an academic editor with relevant expertise and I am writing to let you know that we would like to send your submission out for external peer review.

Once your full submission is complete, your paper will undergo a series of checks in preparation for peer review. After your manuscript has passed the checks it will be sent out for review. To provide the metadata for your submission, please Login to Editorial Manager (https://www.editorialmanager.com/pbiology) within two working days, i.e. by Dec 12 2023 11:59PM.

Kind regards,

Ines

--

Ines Alvarez-Garcia, PhD

Senior Editor

PLOS Biology

---

## [Decision Letter · Decision Letter 1]

23 Feb 2024

Dear Dr Archambault,

Thank you for your patience while your manuscript entitled "Nuclear reassembly defects after mitosis trigger apoptotic and p53-dependent safeguard mechanisms in Drosophila" was peer-reviewed at PLOS Biology. Please also accept my apologies for the delay in providing you with our decision. The manuscript has now been evaluated by the PLOS Biology editors, an Academic Editor with relevant expertise, and by two independent reviewers. 

As you will see, both reviewers find the conclusions novel and interesting, but they also raise several issues that would need to be addressed before we consider the manuscript for publication. Reviewer 1 thinks that parts of the model need to be tested experimentally, that some experiments need to be quantified, and that you should improve the introduction and discussion, adding relevant references that are missing. Reviewer 2 has similar comments, but also asks for confirmation of some of the genetic interactions with molecular markers among other issues.

In light of the reviews, we would like to invite you to revise the work to thoroughly address the reviewers' reports. Given the extent of revision needed, we cannot make a decision about publication until we have seen the revised manuscript and your response to the reviewers' comments. Your revised manuscript is likely to be sent for further evaluation by all or a subset of the reviewers.

**IMPORTANT - SUBMITTING YOUR REVISION**

3. Resubmission Checklist

a) *PLOS Data Policy*

b) *Published Peer Review*

d) *Blurb*

Please also provide a blurb which (if accepted) will be included in our weekly and monthly Electronic Table of Contents, sent out to readers of PLOS Biology, and may be used to promote your article in social media. The blurb should be about 30-40 words long and is subject to editorial changes. It should, without exaggeration, entice people to read your manuscript. It should not be redundant with the title and should not contain acronyms or abbreviations. For examples, view our author guidelines: https://journals.plos.org/plosbiology/s/revising-your-manuscript#loc-blurb

Sincerely,

Ines

--

Ines Alvarez-Garcia, PhD

Senior Editor

PLOS Biology

Reviewers' comments

Rev. 1:

BAF localizes on chromosomes at mitotic exit. This is due to BAF dephosphorylation by PP2A and Ankle2 and is essential for appropriate nuclear envelope reformation. This manuscript from Li et al nicely recapitulates the relationship between Ankle2, BAF and lamin that was described in other systems, and their role in post-mitotic nuclear envelope assembly and tissue integrity in Drosophila. The authors provide an analysis of the consequences of perturbing this process in vivo, in Drosophila imaginal discs. Wing discs with defective nuclear envelopes accumulate dying cells, and lead to small-winged adults. Similar to previous work, they find that lamina defects induce DNA damage, which in turn trigger apoptosis by activation of Chk2. Surprisingly, inhibition of apoptosis does not ameliorate - but exacerbates - the adult phenotype, suggesting that apoptosis promotes tissue integrity. In addition, the authors find that p53-mediated cell cycle arrest also contributes to tissue integrity. This manuscript is clear and well written, and the authors are fair in their claims and cite relevant previous work. Within its scope, this manuscript is interesting and brings a new perspective on nuclear integrity and how this affects epithelial tissue homeostasis. However, there are some major points that I believe should be addressed before accepting this manuscript for publication.

1) Developmental defects in the wings and wing discs were reported for dMAN1 mutant flies (see PMID 18723885). Similarly, lamin depletion using ap-Gal4 causes cell death and defects in adult wings (see PMID 20505245). Unless I missed them, these works should be cited.

2) The authors suggest several times that p53-mediated cell cycle arrest improves cell viability by allowing time for nuclear repair to occur. This is something that could be easily tested. Does depletion of CycE reduce the levels of DNA damage induced by Ankle2 RNAi?

3) As visible from Figure 2A, the scoring of micronuclei should be revised, and the quantifications updated accordingly. The micronuclei highlighted by yellow arrows in the panels showing Ankle2 and BAF RNAi are as large as main nuclei - and, therefore, can't be called micronuclei. Micronuclei are normally small, much smaller than the main nucleus, they may contain one or few mis-segregated chromosomes. They are normally visible as dense DAPI spots, and although there is not a defined size, I believe the scoring used by the authors is not appropriate.

4) DNA damage from abnormal NE causes Chk2-dependent, but p53-independent apoptosis in wing discs where Ankl2 has been knocked down. This suggests than non-apoptotic cell death could be eliminating those cells that are retained as a result of p35 overexpression. These results are interesting, especially in light of what happens when DNA damage is caused by caspase activation. I believe they should be discussed and contextualized with the work of the Milan lab (PMID 37751744).

5) I disagree with the authors' perspective on undead cells in the discussion. The retention of undead cells should not prevent compensatory proliferation. If anything, it should enhance it, because cells with active caspase are retained in the tissue and therefore have the ability of continuing producing growth signals (the authors mention this themselves later). From the results presented, it is clear that retention of abnormal cells is deleterious, because blocking apoptosis, while inhibiting cell death in the disc, is insufficient to rescue the small wing phenotype elicited by abnormal NE in the adult. This could be due to the occurrence of non-apoptotic cell death at later stages of development. Non apoptotic cell death could eliminate those cells that are retained as a result of p35 overexpression. While still in the disc, these undead cells could also exacerbate the phenotype by inducing non-autonomous cell death. These possibilities should be discussed.

6) I suggest reviewing the introduction. Especially the section on micronuclei seems out of place, interrupting an otherwise linear flow of thoughts on nuclear integrity.

Minor comments:

Figures:

Figure 1A, WB: why are there multiple bands for Ankle2? Although the RNAi diminishes the signal from all bands, suggesting that the antibody does indeed recognize Ankle2, it would be reassuring to test the antibody in a mutant or in an overexpression. If mutations are lethal, maybe in flip-out clones with immunofluorescence (if the antibody works in immunofluorescence?)

Figure 1B: [Assuming the labeling is incorrect, and RNAi NT is actually the left panel] Is Flag-BAF enrichment at the NE lost because the NE is compromised, or because BAF is not correctly targeted? It would be helpful to see a staining for lamin. In addition, the cell shown has condensed and fragmented DNA. Can the authors stain with antibodies against cDcp1 to demonstrate that the cell is not dead, or show a cell with less nuclear alterations?

Figure 2A: I think it would be important to differentiate which NE defects are due to Ankle2/BAF/lamin depletion and which defects are a consequence of apoptotic cellular dismantling (partially answered in Figure 5A). Could the authors discriminate between cells that are dead and cells that are alive (maybe with a cDcp1 staining)? Also, some of the cells shown appear much larger than controls. Are defects in cell division occurring in these cells as well? This seems to be the case, given the quantifications on multinucleation (Fig. 2C, although the % reported in the quantification does not match the frequency that can be inferred looking at Fig. 2A). Note that DNA damage could cause defects in cell division, and defects in cell division might exacerbate the observed defects in the NE

Figure 3B: Why do the authors think that lamin depletion has such a small effect on adult wing size compared to Ankle2?

S1A: the data relative to 7 days are very difficult to interpret due to the variability in the loading control (is this due to high level of cell death? Off target of the RNAi? Or just insufficient loading?)

S1B: although I do clearly see an acentric chromosome that will likely reside in a micronucleus, I do not see the micronucleus at 120min (supposedly shown by the yellow arrow, but the yellow arrow just shows the main nucleus at 120min).

S1B: what happens to the second nucleus in RNAi Ankle2? Does it get out of focus or collapses together with the only nucleus visible? Also, why does Lamin remain largely intranuclear in the control, rather than staining the nuclear rim? Is it just a problem of z-projection?

S3: would it be possible to add a panel showing lamin RNAi and the levels of cDcp1?

Introduction:

1) I suggest introducing micronuclei and the mechanisms that lead their formation (most often, mitotic errors). I also recommend including reference to the pivotal work from Kato and Sandberg (PMID 5635016) and the Pellman lab (PMID 22258507) and explaining why DNA isolation in micronuclei is associated with formation of DNA damage (e.g. incomplete replication and exposure of DNA to the cytoplasm). In addition, I believe the paragraph does not flow well with the surrounding text. Could the authors consider a different placement?

2) Regarding whether micronuclei can trigger apoptosis, please note that most evidence supporting apoptosis after micronucleation relies on treatment with pleiotropic, cytotoxic drugs. I am not sure whether micronucleation per se was reported to function as an apoptotic stimulus, and whether a clear-cut difference was made between the apoptotic potential of micronucleation vs aneuploidization.

3) I advise against the use of the acronym NR (for Nuclear Reassembly). While other acronyms are common (like NE for Nuclear Envelope), this is unusual and does not help readability of the manuscript

Results:

1) Definition of D-mel abbreviation is missing (should be indicated when first mentioned)

2) "Depletion of Ankle2, BAF or Lamin also caused delocalization of Nuclear Pore Complex (NPC) proteins and Otefin (an inner nuclear membrane protein with a LEM domain that interacts with BAF) from the NE (Fig S3D)" - I believe the authors refer to Fig S3C. I find it very surprising that BAF knock down causes a milder phenotype than Ankle2, if Ankle2 effects are mostly due to its role in BAF dephosphorylation. Is this due to the use of a weaker RNAi line?

3) Although it is clear why overexpression of GFP-BAF3A rescues Ankle2 depletion, it is less clear why any rescue should occur by overexpressing GFP-BAFWT. Could the authors elaborate? Do they expect a fraction of the WT protein to escape phosphorylation?

4) Figure S4D: what is the explanation for lack of statistical significance in the TUNEL assay between control and BAF RNAi?

Rev. 2:

In this manuscript, Li et al. dissect the molecular mechanisms promoting nuclear envelope reformation (NER) during mitotic exit and investigate the cellular and physiological consequences of nuclear reformation defects on tissue development in Drosophila melanogaster.

In the first part of the manuscript, they provide evidence that Anlke2/Lem4 promotes BAF dephosphorylation during mitotic exit to induce NER. Inactivation of Ankle2/Lem4 prevents BAF recruitment to the chromosomes, and leads to major NER defects. The inactivation of BAF itself recapitulates this phenotype. These observations are consistent with previous observations in mammalian cells and in C. elegans embryos.

In the second part, using the Drosophila wing as a model system, they evaluate the physiological consequences of nuclear reformation defects on tissue development. They nicely show that NER defects lead to the development of smaller wings, and that this phenotype can be enhanced or suppressed using mutations aggravating or suppressing (respectively) NER defects. They further show that NER defects trigger an apoptotic response, which is required for the wing tissue to recover from the NER- defect-induced damage. Activation of apoptosis in response to NER defects involves the activation of the DNA damage response kinase Chk2, but is independent of p53 activation. Nevertheless, in response to NER defects, p53 promotes a cell cycle arrest in G1 via the ubiquitin-mediated degradation of Cyclin E. Both the apoptotic response and the cell cycle arrest are beneficial to wing tissue development.

Overall, this is an interesting manuscript that nicely links cellular defects to tissue development. However, I found several parts confusing and needing clarification.

Specific points:

- The Drosophila genome possesses two genes encoding lamins, Lamin C and lamin Dm0 but it is unclear which lamin the authors are monitoring. When they do lamin RNAi, what does it mean? Do they inactivate only one lamin or both? This information is critical to understand the results. Furthermore, the authors monitor GFP-lamin localization, is this fusion protein fully functional?

- Page 8, at the beginning of the paragraph "Nuclear reassembly defects trigger apoptosis": the authors should state more clearly that they are monitoring apoptosis in the fly wings. A non-Drosophila reader would not really know what tissue is being looked at. This information eventually appears only on line 12.

- The statistical analyses are questionable and some information is missing (n= sample size and N=number of independent experiments performed). On several graphs, the authors add a p value <0.0001 (****) but just by looking at the data points between conditions, one can wonder how an unpaired t-test can give such a low p value. For instance, Fig2B where the authors compare the effect of lamin-, BAF- or Ankl2-RNAi on nuclear solidity to the control. In all cases, comparison to the control gives a p value <0.0001, and yet Ankle2 RNAi appears to have a much stronger effect. The same comment applies to Figures 3B, 7A, S1C. Maybe the authors could double-check their statistical analysis.

- Please use the same nomenclature for the RNAi between the figures and the text.

- All the interactions in Figure 7 are entirely based on genetic interactions, which are nicely designed and conducted, but confirming these interactions using molecular markers would strengthen the conclusions. For instance, demonstrating that introduction of an ago mutation directly causes Cyclin E accumulation in their system would consolidate the conclusions. Likewise, presenting molecular evidence directly showing that p53 is activated in response to NER defects would be desirable.

Figure 1:

Figure 1B: RNAi Ankle 2 and RNAi NT panels are inverted. Also, the authors should write "control" instead of "NT" because NT is not self-explanatory.

Figure 1B and Figure 1D: the authors mentioned that they quantified BAF levels at the NE but they did not use any nuclear envelope marker for the quantification! How did they quantify the NE levels of BAF in these conditions? This information is not provided in the Materials and Methods section.

Figure 1B: Why is there less FLAG-BAF signal in the Ankle RNAi compared to the RNAi Ctrl? This panel is confusing when we compare it to the first panel presented Figure 1E (-43 min). The authors should clarify this point.

Figure 1E: The authors should present pics taken with the same timing intervals and maybe also include an extra panel, to show that even after 113 min, BAF is still not recruited to the NE after Ankle2 inactivation. Some points are missing in the image (the highest GFP-BAF level is at 7 min), which are required for the interpretation of the graph (GFP-BAF recruitment to reassembling nuclei).

Figure 1E: Add a panel showing BAF 3A localization during mitosis.

Figure 1E: As mentioned above, at -44.3 min (interphase), GFP-BAF presents the exact same localization in the NT and RNAi treated cells, whereas in Figure 1B in the Ankle RNAi the GFP-BAF is not in the NE! This is confusing and should be clarified.

Figure 1F: As the authors mentioned previous work performed on C. elegans BAF. It would be useful to include it in the alignment.

Figure 1G: This western blot would be more convincing if the authors include a loading control and quantification.

Figure 2: Again, please mention which lamin is monitored in panel A. In Panel B, explain in more detail the quantification (define "solidity"). In addition, this graph should start at "0". Panel D: Explain what is considered as "abnormal Lamin", and how was it quantified?

Figure 4, panel B: There are clearly more Dcp-1 dots in Ankle2 RNAi compared to Lamin RNAi on the presented images but the graph in Figure 4C shows the opposite. This is confusing. Panel E: add Dcp-1/Tunel for the images presented on top and Tunel/DNA for the images at the bottom.

Figure S3B: Compare TioOE with BAF RNAi #1 (40D) because this is the real control and not the No RNAi. Verify the statistical analysis p value.

Figure S3C: Which antibody was used to detect the NPCs?

---

## [Decision Letter · Decision Letter 2]

1 Aug 2024

Dear Dr Archambault,

Thank you for the submission of your revised Research Article entitled "Nuclear reassembly defects after mitosis trigger apoptotic and p53-dependent safeguard mechanisms in Drosophila" for publication in PLOS Biology. On behalf of my colleagues and the Academic Editor, Renata Basto, I am delighted to let you know that we can in principle accept your manuscript for publication, provided you address any remaining formatting and reporting issues. These will be detailed in an email you should receive within 2-3 business days from our colleagues in the journal operations team; no action is required from you until then. Please note that we will not be able to formally accept your manuscript and schedule it for publication until you have completed any requested changes.

PRESS

Sincerely, 

Ines

--

Ines Alvarez-Garcia, PhD

Senior Editor

PLOS Biology
